

# Effects of Urbanization on Regional Meteorology and Air Quality in Southern California

## Yun Li[1], Jiachen Zhang[1], David J. Sailor[2], George A. Ban-Weiss[1]

[1]Department of Civil and Environmental Engineering, University of Southern California, Los Angeles, 90007, USA

[2]School of Geographical Science and Urban Planning, Arizona State University, Tempe, 85281, USA

*Correspondence to*: George Ban-Weiss (banweiss@usc.edu)

## Abstract

Urbanization has a profound influence on regional meteorology and air quality in megapolitan Southern California. The influence of urbanization on meteorology is driven by changes in land surface physical properties and land surface processes. These changes in meteorology in turn influence air quality by changing temperature-dependent chemical reactions and emissions, gas-particle phase partitioning, and ventilation of pollutants. In this study we characterize the influence of historical urbanization from before human settlement to present-day on meteorology and air quality in Southern California using the Weather Research and Forecasting Model coupled to chemistry and the single-layer urban canopy model (WRF/Chem-UCM). We assume identical anthropogenic emissions for the simulations carried out, and thus focus on the effect of changes in land surface physical properties and land surface processes on air quality. Historical urbanization has led to daytime air temperature decreases of up to 1.4 K, and evening temperature increases of up to 1.7 K. Ventilation of air in the LA basin has decreased up to 36.6% during daytime and increased up to 27.0% during nighttime. These changes in meteorology are mainly attributable to higher evaporative fluxes from irrigation, higher thermal inertia from irrigation and building materials, and increased surface roughness from buildings. Changes in ventilation drive changes in hourly NOx concentrations with increases of up to 2.7 ppb

during daytime and decreases of up to 4.7 ppb at night. Hourly $O_3$ concentrations decrease by up to 0.94

ppb in the morning, and increase by up to 5.6 ppb at other times of day. Changes in $O_3$ concentrations

are driven by the competing effects of changes in ventilation and precursor NOx concentrations. $PM_{2.5}$

concentrations show slight increases during the day, and decreases of up to 2.5 µg/$m^3$ at night.

Processes drivers for changes in $PM_{2.5}$ include modifications to atmospheric ventilation, and

temperature, which impacts gas-particle phase partitioning for semi-volatile compounds and chemical

reactions. Understanding processes drivers for how land surface changes effect regional meteorology

and air quality is crucial for decision making on urban planning in megapolitan Southern California to

achieve regional climate adaptation and air quality improvements.

## 1. Introduction

The world has been undergoing accelerated urbanization since the industrial revolution in the 19[th]

Century (Grimm et al., 2008; Seto et al., 2012). Urbanization leads to profound human modification of

the land surface and its associated physical properties such as roughness, thermal inertia, and albedo

(Fan et al., 2017), and land surface processes like irrigation (Vahmani and Hogue, 2014). These changes

in land surface physical properties and processes alter corresponding surface-atmosphere coupling

including exchange of water, momentum and energy in urbanized regions (Vahmani and Ban-Weiss,

2016a; Li et al., 2017), which exerts an important influence on regional meteorology and air quality

(Vahmani et al., 2016; Civerolo et al., 2007).

Land surface modifications from urbanization drive changes in urban meteorological variables such

as temperature, wind speed and planetary boundary layer (PBL) height, which result in urban – rural

differences. Differences in surface temperature and near surface air temperature have been widely

studied for decades. The urban heat island (UHI) effect, a phenomenon in which temperatures within an



urban area are higher than surrounding rural areas (Oke, 1982), has been extensively studied using models and observations for a great number of urban regions (Rizwan et al., 2008; Peng et al., 2012; Stewart and Oke, 2012). A contrary phenomenon, namely the urban cool island (UCI), under which urban temperatures are lower than surrounding rural temperatures, has also been investigated recently in some studies (Carnahan and Larson, 1990; Theeuwes et al., 2015; Kumar et al., 2017). Urban – rural

contrast in temperature (i.e. both UHI and UCI) is mainly attributable to differences in thermal properties and energy fluxes due to heterogeneous land surface properties. For urban areas, buildings and roads (i.e., impervious surfaces) are generally made from manufactured materials (e.g., asphalt concrete) with low albedo and thus high solar absorptivity (Wang et al., 2017). These materials also have high thermal inertia, which can lead to reductions in diurnal temperature range due to heat storage

and consequent temperature reductions during the day and heat release and consequent temperature increases at night (Hardin and Vanos, 2018). Street canyons, which we refer to as the U-shaped region between buildings, can trap longwave energy fluxes within the canyon because of reductions of sky-view factors (Qiao et al., 2013). On the other hand, shading in street canyons during the day can reduce absorption of shortwave radiation (Carnahan and Larson, 1990; Kusaka et al., 2001). Pervious

surfaces within urban areas such as irrigated urban parks and lawns can lead to the urban oasis effect in which evaporative cooling occurs due to increases in evapotranspiration. In addition, soil thermal properties depend on their water content, which ultimately affects ground heat fluxes and thus surface and air temperatures. Land surface properties in surrounding rural areas can also affect urban – rural differences in temperature (Imhoff et al., 2010; Peng et al., 2012; Zhao et al., 2014). Urban regions built

in semi-arid or arid surroundings tend to have a weak UHI or even a UCI, whereas those built in moist regions tend to have a larger UHI. Lastly, factors such as anthropogenic heat and atmospheric aerosol burdens can play an important role in urban heat/cool island formation in some regions (Oke, 1982; Wang et al., 2017).



Urbanization can also cause differences between urban and rural areas for meteorological variables

other than surface and air temperatures. Changes in regional near-surface wind speed and direction can

occur in urban areas because of spatially varying modifications in surface roughness (Xu et al., 2006;

Vahmani et al., 2016). Changes in near surface winds in coastal urban areas can also be affected by

modifying land-sea temperature contrast (Vahmani et al., 2016). The formation of the PBL is dependent

on the magnitude of turbulent kinetic energy (TKE) (Garratt, 1994). Higher (lower) TKE will lead to

deeper (shallower) PBLs. During daytime, the magnitude of TKE is driven by buoyancy production

contributed mainly by sensible heat flux (with clear skies); at night, TKE is driven by shear production

associated with variance in wind speed. Thus, temperature and surface roughness play an important role

on the depth of the PBL during daytime and nighttime, respectively. Lastly, changes in relative humidity,

precipitation, and other meteorological variables due to land surface changes can also be significant in

some regions (Burian and Shepherd, 2005; Georgescu et al., 2014).

Changes in meteorological conditions from urbanization can influence concentrations of air

pollutants including oxides of nitrogen (NOx), ozone ($O_3$) and fine particulate matter ($PM_{2.5}$). NOx and

$O_3$ pollution are major public health concerns in megapolitan regions (Lippmann, 1989). $PM_{2.5}$ reduces

visibility, exerts adverse health effects, and alters regional climate via direct and indirect effects

(Charlson et al., 1992; Pope and Dockery, 2006; Boucher et al., 2013). Changes in air pollutant

concentrations due to meteorological changes occur via altered emissions, weather-dependent chemical

reactions, gas-particle phase partitioning of semi-volatile species, pollutant dispersion, and deposition.

Variations in air temperatures together with vegetation types affect the production of biogenic volatile

organic compounds (BVOCs), which are important precursors for ground-level $O_3$ and secondary

organic aerosols (SOA) (Guenther et al., 2006). Gas-phase chemical reactions that form secondary

pollutants are also temperature-dependent. Higher (lower) air temperatures in general lead to higher



(lower) photolysis reaction rates and atmospheric oxidation rates, which enhance the production of tropospheric $O_3$, secondary inorganic aerosols (e.g. nitrate, sulfate, and ammonium aerosols) and SOA (Aw and Kleeman, 2003; Hassan et al., 2013). In addition, concentrations of semi-volatile compounds are affected by equilibrium vapor pressure under various temperature conditions (Pankow, 1997; Ackermann et al., 1998). Higher (lower) temperatures favor phase-partitioning to the gas (particle) phase. Ventilation, which is the combined effect of vertical mixing and horizontal dispersion, can also influence pollutant concentrations (Epstein et al., 2017). Higher (lower) ventilation rates lead to lower (higher) pollutant concentrations especially in coastal cities like Los Angeles where upwind air under typical meteorological conditions is clean relative to urban air. Lastly, changes in surface roughness may affect loss of pollutants via surface deposition, which in turn alters air pollutant concentrations (Abdul-Wahab et al., 2005).

A number of previous studies have investigated the impacts of land surface changes on regional meteorology in a variety of urban regions around the world (Kalnay and Cai, 2003; Burian and Shepherd, 2005; Zhang et al., 2010). However, limited studies have quantified the impact of land surface changes on regional air quality, and most of these studies have focused on changes in surface $O_3$ concentrations. Civerolo et al. (2007) estimated that land-use changes via urban expansion in New York City can cause increases in near-surface air temperature of 0.6 ℃ as well as increases in episode-maximum 8h $O_3$ concentrations of 6 ppb. Jiang et al. (2008) focused on the Houston, Texas area, and found similar relationships between urban expansion, near-surface air temperatures, and $O_3$. Nevertheless, only a few studies have included changes in $PM_{2.5}$ concentrations. Tao et al. (2015) simulated that spatially averaged surface $O_3$ concentrations slightly increased (+0.1 ppb) in eastern China due to urbanization, whereas $PM_{2.5}$ concentrations decreased by –5.4 $\mu g/m^3$ at the near surface. Chen et al. (2018) studied urbanization in Beijing, and found that modification of rural to urban land





surfaces has led to increases in near-surface air temperature and PBL height, which in turn led to increases (+9.5 ppb) in surface $O_3$ concentrations and decreases (–16.6 µg/m$^3$) in PM$_{2.5}$ concentrations. However, past studies that investigate interactions between land surface changes and changes in meteorology and air quality generally do not identify the major processes driving these interactions. They also do not resolve the wide heterogeneity of urban land surface properties, with most studies

assuming that urban properties are homogenous throughout the city. In addition, only few studies investigate interactions between land surface changes and air quality for the Southern California region (Taha, 2015; Epstein et al., 2017; Zhang et al., 2018b), which has among the worst air quality in the United States (American Lung Association, 2012).

With advances in real-world land surface datasets from satellites, recent modeling studies on

land-atmosphere interactions are able to resolve heterogeneous land surface properties and thus better capture urban meteorology, enabling modeling studies that more accurately quantify changes in regional meteorology due to land surface modification. By combining satellite-retrieved high-resolution land surface data with the Weather Research and Forecasting Model coupled to the Single-layer Urban Canopy Model (WRF/UCM), simulations reported in Vahmani and Ban-Weiss (2016a) show improved

model performance (i.e. compared to observations) for meteorology in Southern California compared to the default model, which assumes that urban regions have homogeneous urban land cover. A follow-up study, Vahmani et al. (2016), suggested that historical urbanization has altered regional meteorology (e.g., near surface air temperatures and wind flows) in Southern California mainly because of urban irrigation, and changes in land surface thermal properties and roughness. While historical urbanization

and its associated impacts on meteorology has the potential to cause important changes in air pollutant concentrations in Southern California, this is never been investigated in past work.

Therefore, this study aims to characterize the influence of historical urbanization on urban





meteorology and air quality in Southern California by comparing a "Present-day" scenario with current

urban land surface properties and land surface processes to a "Nonurban" scenario assuming land

surface distributions prior to human perturbation. To achieve this goal, we adopt a state-of-the-science

regional climate-air quality model, the Weather Research and Forecasting Model coupled to chemistry

and the Single-layer Urban Canopy Model (WRF/Chem-UCM), and incorporate high-resolution

heterogeneity in urban surface properties and processes to predict regional weather and pollutant

concentrations. We assess the response of regional meteorology and air quality to individual changes in

land surface properties and processes to determine driving factors on atmospheric changes.

## 2. Methodology and Data

### 2.1 Model Description and Configuration

WRF/Chem v3.7 is used in this study to simulate meteorological fields and atmospheric chemistry.

WRF/Chem is a state-of-the-science nonhydrostatic mesoscale numerical meteorological model that

facilitates "online" simulation of processes relevant to atmospheric chemistry including pollutant

emissions, gas and particle phase chemistry, transport and mixing, and deposition (Grell et al., 2005). In

this study, we couple WRF/Chem to the urban canopy model (UCM) that resolves land-atmosphere

exchange of water, momentum, and energy for impervious surfaces in urban areas (Kusaka et al., 2001;

Chen et al., 2011; Yang et al., 2015). The UCM parameterizes the effects of urban geometry on energy

fluxes from urban facets (i.e., roofs, walls, and roads) and wind profiles within canyons (Kusaka et al.,

2012). We account for the effect of anthropogenic heat on urban climate by adopting the default diurnal

profile in the UCM. Physics schemes included in our model configuration are the Lin cloud

microphysics scheme (Lin et al., 1983), the RRTM longwave radiation scheme (Mlawer et al., 1997),

the Goddard shortwave radiation scheme (Chou and Suarez, 1999), the YSU boundary layer scheme



(Hong et al., 2006), the MM5 similarity surface layer scheme (Dyer and Hicks, 1970; Paulson, 1970), the Grell 3D ensemble cumulus cloud scheme (Grell & Dévényi, 2002), and the unified Noah land surface model (Chen et al, 2001). Chemistry schemes include the TUV photolysis scheme (Madronich, 1987), RACM-ESRL gas phase chemistry (Kim et al., 2009; Stockwell et al., 1997), and MADE/VBS aerosols scheme (Ackermann et al., 1998; Ahmadov et al., 2012).

All model simulations are carried out from June 28[th], 2100 UTC (June 28[th], 1300 PST) to July 8[th], 0700 UTC (July 7[th], 2300 PST), 2012 using the North American Regional Reanalysis (NARR) dataset as initial and boundary meteorological conditions (Mesinger et al., 2006). This simulation period is chosen as representative of typical summer days in Southern California. Hourly model output from July 1[st], 0000 PST to 7[th], 2300 PST is used for analysis, and simulation results prior to July 1[st], 0000 PST are

discarded as spin up. Figure 1a shows the three two-way nested domains with horizontal resolutions of 18 km, 6 km and 2 km, respectively, centered at 33.9°N, 118.14°W. Only the innermost domain (141 ×129 grid cells), which encapsulates the Los Angeles and San Diego metropolitan regions, is used for analysis. All three domains consist of 29 unequally spaced layers in the vertical from the ground to 100 hPa.

**2.2 Land Surface Property Characterization and Process Parameterization**

    One important aspect of accurately simulating meteorology and air quality is to properly characterize land surface – atmosphere interactions (Vahmani and Ban-Weiss, 2016a; Li et al., 2017). In addition, accurately quantifying the climate and air quality impacts of historical urbanization requires a realistic portrayal of current land cover in the urban area (Vahmani et al., 2016). For both of these

reasons, we update the default WRF/Chem to include a real-world representation of land surface physical properties and processes.



In this study, we use the (30 m resolution) 33-category National Land Cover Database (NLCD) for the year 2006 for all three model domains. NLCD differentiates three urban types including low-intensity residential, high-intensity residential, and industrial/commercial (shown in Figure 1b) (Fry et al., 2011). In the model, each of these three types can have unique urban physical properties such as building morphology, albedo, and thermal properties for each facet. We adopt the grid-cell specific National Urban Database and Access Portal Tool (NUDAPT) where available in the innermost domain for building morphology including average building heights, road widths, and roof widths (Ching et al., 2009). Where NUDAPT data are unavailable, we use average building and road morphology for three urban categories from the Los Angeles Region Imagery Acquisition Consortium (LARIAC). Details on the generation of averaged urban morphology parameters from real-world GIS datasets can be found in Zhang et al. (2018a). Note that the original gaseous dry deposition code based on Wesely (1989) is only compatible with the default 24-category U.S. Geological Survey (USGS) global land cover map. We therefore modify the code according to Fallmann et al. (2016), which assumes that the three urban types in the 33-category system have input resistances that are the same as the urban type for the 24-category system. In addition, impervious fractions (i.e. the fraction of each cell covered by impervious surfaces) for each of the three urban categories in the innermost domain are from the NLCD impervious surface data (Wickham et al., 2013).

Land surface properties including albedo, green vegetation fraction (GVF), and leaf area index (LAI) are important for accurately predicting absorption and reflection of solar radiation and evaporative fluxes in urban areas (Vahmani and Ban-Weiss, 2016a). To resolve high-resolution real-world heterogeneity in these land surface properties, the simulations performed in this study use satellite-retrieved real-time albedo, GVF, and LAI for the innermost domain. Input data compatible with WRF are regridded horizontally using albedo, GVF, and LAI maps generated based on MODIS

reflectance (MCD43A4), vegetation indices (MOD13A3), and fraction of photosynthetically active

radiation (MCD15A3) products, respectively. Raw data are available from the USGS National Center

for Earth Resource Observations and Science website at http://earthexplorer.usgs.gov. A detailed

description on the implementation of MODIS-retrieved land surface properties for WRF can be found in

Vahmani and Ban-Weiss (2016a). Our previous research has shown that the model enhancements

described here reduce model biases in surface and near-surface air temperatures (relative to ground and

satellite observations) for urban regions in southern California. In particular, the root-mean-square-error

for nighttime near-surface air temperature has been narrowed from 3.8 to 1.9 °C.

Resolving urban irrigation is also of great significance for accurately predicting latent heat fluxes

and temperatures within Los Angeles. Here we use an irrigation module developed by Vahmani and

Hogue (2014), which assumes irrigation occurs three times a week at 2100 PST. This model was tuned

to match observations of evapotranspiration in the Los Angeles area. Details on the implementation of

this irrigation module and its evaluation with observations can be found in Vahmani and Hogue (2014).

Note that we do not use the default irrigation module available in the single layer canopy model in

WRF/UCM v3.7, which assumes daily irrigation at 2100 PST in summertime, because (1) the irrigation

module of Vahmani and Hogue (2014) was already evaluated and tuned for Southern California, and (2)

we strive to maintain consistency with our previous related studies.

## 2.3 Emission Inventories

Producing accurate air quality predictions also relies on using emission inventories that capture

real-world emissions. We adopt year 2012 anthropogenic emissions from the California Air Resource

Board (CARB) for the two outer domains (CARB, 2017) where data are available (i.e. within

California), and from South Coast Air Quality Management District (SCAQMD) for the innermost

domain (SCAQMD, 2017). For areas within the two outer domains that are outside California, we use



the U.S. Environmental Protection Agency (EPA) National Emissions Inventory (NEI) for 2011 that is available with the standard WRF/Chem model (U.S. EPA, 2014). CARB and SCAQMD emission

inventories as provided have 4 km spatial resolution, with 18 and 11 layers in the vertical from the ground to 100 hPa, respectively. We regridded these inventories in the horizontal and vertical to match the grids of our modeling domains. Note that the aforementioned emission inventories use chemical speciation from the SAPRC chemical mechanism (Carter, 2003), and thus we have converted species to align with the RACM-ESRL and MADE/VBS mechanisms, both of which use RADM2 (Regional Acid

Deposition Model) speciation (Stockwell et al., 1990). The conversion uses species and weighting factors from the emiss_v04.F script that is distributed with NEI emissions for WRF/Chem modeling. (The original script is available at:    ftp://aftp.fsl.noaa.gov/divisions/taq.) More details on re-speciating the emissions datasets are presented in the supplemental information (Table S1). For online calculation of biogenic volatile organic emissions we adopt the Model of Emissions of Gases and Aerosols from

Nature (MEGAN) (Guenther et al., 2006). The default LAI in MEGAN is substituted with the satellite-retrieved LAI for better quantification of biogenic emissions. Note that we have turned on online calculation of sea salt emissions, but turned off that of dust emissions (both available with default WRF).

## 2.4 Meteorology and Air Pollutant Observations

To facilitate model evaluation, we obtain hourly near-surface air temperature observations, hourly ground-level $O_3$ and daily $PM_{2.5}$ observations within our simulation period. Near-surface air temperature data are gathered from 12 stations from the California Irrigation Management Information System (CIMIS). Air pollutant observations are from the Air Quality System (AQS), which is maintained by the U.S. EPA. Ozone ($PM_{2.5}$) data from 33 (27) air quality monitoring stations are

collected representing Los Angeles, Orange, Riverside and San Bernardino Counties. The locations of



monitoring stations are shown in Figure S5. Among the 27 monitoring stations where $PM_{2.5}$

observations are available, daily $PM_{2.5}$ concentrations from gravimetric analysis can be directly obtained

from 20 stations, while hourly observations acquired using a Beta Attenuation Monitoring (BAM) are

obtained from 15 stations. Hourly $PM_{2.5}$ observations at each station are temporally averaged to obtain

daily $PM_{2.5}$ values.

## 2.5 Simulation Scenarios

To investigate the effects of land surface changes via historical urbanization on regional

meteorology and air quality in Southern California, we carry out two simulations, which we refer to as

the "Present-day" scenario and "Nonurban" scenario. The two scenarios differ only by the assumed land

surface properties and processes, which are shown in Figure 2. The Present-day scenario assumes the

land cover (Figure 1b) and irrigation of current for Southern California (described in Section 2.2).

Urban morphology from NUDAPT and LARIAC, and MODIS-retrieved albedo, GVF and LAI are used

in this scenario. To help explain the impact of urbanization without the addition of irrigation, a

supplemental simulation, which we refer to as "Present-day No-irrigation", is also carried out; this

simulation is identical to "Present-day" but assumes that there is no irrigation. For the Nonurban

scenario, we assume natural land cover prior to human perturbation, and replace all urban grid cells

with "shrubs" (Figure 1c). We modify MODIS-retrieved albedo, GVF and LAI in these areas based on

properties for shrub lands surrounding urban regions in the Present-day scenario. A detailed explanation

on this method can be found in Vahmani et al. (2016). The spatial pattern of land surface properties in

both "Present-day" and "Nonurban" scenarios are shown in Figure S6. Note that all three

aforementioned scenarios adopt identical anthropogenic emission inventories described in Section 2.3

since the focus of this study is to investigate urbanization effects on meteorology and air pollutant

concentrations. (Biogenic emissions do change for the scenarios due to changes in land surface

properties (e.g., vegetation type and LAI) and meteorology (e.g., temperature).)

## 2.6 Uncertainties


Note that the results reported in this paper are based on model simulations and are thus dependent on how accurately the regional climate/chemistry model characterizes the climate/chemistry system (e.g., meteorology, surface-atmosphere coupling, and atmospheric chemical reactions). Results may be dependent on model configuration (e.g., physical and chemical schemes), land surface characterizations (e.g., satellite data from MODIS, or default dataset available in WRF) and emission inventories (e.g., anthropogenic emission inventories from CARB, SCAQMD or NEI). In addition, since irrigation is not included in the Nonurban scenario, simulated meteorology in the Nonurban scenario are dependent on assumed soil moisture initial conditions. In this study, we adopt the initial soil moisture conditions from Vahmani et al. (2016) for consistency with our previous work. Soil moisture initial conditions are based on values from six-month simulations without irrigation (Vahmani and Ban-Weiss, 2016b).



# 3. Results and Discussion

## 3.1 Evaluation of Simulated Meteorology and Air Pollutant Concentrations

In this section, we focus on the predicative capability of the model for simulated near-surface air temperature, $O_3$ and total $PM_{2.5}$ concentrations (including sea salt, but excluding dust) for the Present-day scenario. Note that for the evaluation of $PM_{2.5}$ concentrations we include only observations from daily (gravimetric) measurements in this section. The comparison between modeled $PM_{2.5}$ concentrations versus daily averaged observations derived from hourly BAM measurements is discussed in the supplemental information section S1. As shown in Figure 3, predictions of near-surface air temperatures, $O_3$ and $PM_{2.5}$ concentrations show good fit with observations at low values that occur


with high occurrence frequency. However, observed near-surface air temperatures, $O_3$ and $PM_{2.5}$ concentrations are underestimated by the model at higher values that occur with lower occurrence frequency. The underestimation of $PM_{2.5}$ concentrations may be in part due to not accounting for naturally occurring dust emissions in model simulations. Table 1 shows four statistical metrics for model evaluation, including mean bias (MB) and normalized mean bias (NMB) for the quantification of

bias, and mean error (ME) and root mean square error (RMSE) for the quantification of error. The statistical results indicate that while model simulations underestimate near-surface air temperature, $O_3$ and $PM_{2.5}$ concentrations by 0.3%, 22% and 31%, respectively, their performance on capturing observations are acceptable.

## 3.2 Effects of Urbanization on Air Temperature and Ventilation Coefficient

The effects of land surface changes via urbanization in Southern California on air temperature and ventilation coefficient are discussed in this section. Air temperatures are reported for the lowest atmosphere model layer rather than the default diagnostic 2m (near-surface) air temperature variable to be consistent with reported air pollutant concentrations shown in later sections. (The chemistry code makes use of grid cell air temperature and does not use 2m air temperature.) Ventilation coefficient is

calculated as the product of PBL height and the average wind speed within the PBL, and thus considers the combined effects of vertical and horizontal mixing, and indicates the ability of the atmosphere to disperse air pollutants (Ashrafi et al., 2009).

### 3.2.1 Spatial average temperature change

As shown in Figure 4a, urbanization in Southern California has in general led to urban temperature

reductions during daytime from 7 PST to 16 PST, and urban temperature increases during other times of day. The largest spatially averaged temperature reduction occurs at 10 PST ($\Delta T = -1.4$ K), whereas the



largest temperature increase occurs at 20 PST (+1.7 K). Additionally, urbanization led to spatially averaged reduction in diurnal temperature range by 1.5 K. Spatially averaged urban temperature reduction during morning (i.e., defined here and in the following sections as 7:00 – 12:00 PST) and afternoon (i.e., 12:00 – 19:00 PST) are –0.9 K and –0.3 K, respectively. At nighttime (i.e., 19:00 – 7:00 PST), the spatially averaged temperature increase is +1.1 K.

### 3.2.2 Spatial distributions of temperature change

During the morning, temperature reductions are larger in regions further away from the sea (e.g., San Fernando Valley and Riverside County) than coastal regions (e.g., west Los Angeles and Orange County) (Figure 5a). (Note that regions that are frequently mentioned in this study are in Figure 2a.) Spatial patterns in the afternoon are similar to morning, with the exception that coastal regions experience temperature increases (as opposed to decreases) of up to +0.82 K (Figure 5b). During nighttime, temperature increases spread throughout urban regions, and are generally larger in the inland regions of the basin relative to coastal regions (Figure 5c).

### 3.2.3 Processes driving daytime changes

The temporal and spatial patterns of air temperature changes suggest that the climate response to urbanization during daytime is mainly associated with the competition between (a) temperature reductions from increased evapotranspiration and thermal inertia from urban irrigation, and (b) temperature increases from decreased onshore sea breezes (Figure S7a, b). Decreases in the onshore sea breeze are primarily caused by increased roughness lengths from urbanization. (Note that the onshore sea breeze decreases in strength despite higher temperatures in the coastal region of Los Angeles, which would tend to increase the land-sea temperature contrast and thus be expected to increase the sea breeze strength.) Inland regions show larger temperature reductions relative to coastal because they have lower



urban fractions (Figure S6a), and thus higher pervious fractions. Since irrigation increases soil moisture

in the pervious fraction of the grid cell in this model, irrigation will have a larger influence on grid cell

averaged latent heat fluxes (Figure S8) and thermal inertia when pervious fractions are higher. The

inland regions are also less affected by changes in the sea breeze relative to coastal regions since they

are (a) farther from the ocean, and (b) experience smaller increases in roughness length. Roughness

length effects on the sea breeze are especially important in the afternoon when baseline wind speeds are

generally highest in the Los Angeles basin. Thus, the afternoon temperature increases simulated in the

coastal region occur because temperature increases from reductions in the afternoon onshore flows

dominate over temperature decreases from increased evapotranspiration. In addition, increases in

thermal inertia caused by use of manmade materials (e.g., pavements and buildings) and shading effects

within urban canopies can contribute to simulated temperature reductions during the morning. Please

see the supplemental information section S2 for the additional simulation (Present-day No-irrigation

scenario) carried out to identify the influence of urbanization but without changing irrigation relative to

the Nonurban scenario (i.e., with no irrigation).

Note that changes in air temperature during daytime shown here disagree with Vahmani et al.

(2016). While our study detects daytime temperature reductions due to urbanization, Vahmani et al.

(2016) suggests daytime warming. After detailed comparison of the simulations in our study versus

Vahmani et al. (2016), we find that the differences are mainly associated with UCM configuration. First,

our study accounts for shadow effects in urban canopies, whereas Vahmani et al. (2016) assumes no

shadow effects. (We note here that the default version of the UCM has the shadow model turned off.

The boolean SHADOW variables in module_sf_urban.F needs to be manually switched to true to enable

the shadow model calculations. With the shadow model turned off, all shortwave radiation within the

urban canopy is assumed diffuse.) We suggest that it is important to include the effects of building

morphology on shadows within the canopy, and to track direct and diffuse radiation separately, and therefore perform simulations in this study with the shadow model on. Second, our study uses model default calculations of surface temperature for the impervious portion of urban grid cells, whereas Vahmani et al. applied the alternative calculation proposed by Li and Bou-Zeid, 2014. Li and Bou-Zeid, 2014 intended the alternate surface temperature calculation to be performed as a post-processing step rather than during runtime.

### 3.2.4 Processes driving nighttime changes

The temporal and spatial patterns of air temperature changes suggest that the climate response to urbanization during nighttime is driven by the competition between (a) temperature increases from increasing upward ground heat fluxes, and (b) temperature reductions from increasing PBL heights. Increased soil moisture (from irrigation) and use of man-made materials leads to higher thermal inertia of the ground; this in turn leads to increased heat storage during the day and higher upward ground heat fluxes and thus surface temperatures at night. The magnitude of air temperature change during nighttime is also related to the magnitude of PBL height changes; increasing PBL heights can counteract warming driven by higher upward ground heat flux. Greater increases in PBL heights will lead to increasingly diminished air heating rates, and thus smaller air temperature increases. Changes in PBL heights are associated with surface roughness changes since shear production dominants TKE at night. Coastal (inland) regions have larger (smaller) increase in roughness length (Figure 2e), which leads to larger (smaller) increases in PBL heights, and thus smaller (greater) increase in air temperature.

### 3.2.5 Temporal and spatial patterns of ventilation changes and process drivers

Changes in ventilation coefficient show a similar temporal pattern as air temperature (Figure 4b); values decrease by up to –36.6% (equivalent to –826 m$^2$/s, at 10 PST) during daytime, and increase up

to +27.0% (equivalent to +77 m$^2$/s, at 23 PST) during nighttime, due to urbanization. Absolute

reductions in ventilation coefficient are more noticeable in the afternoon than in the morning; the

spatially averaged decreases are –726 m$^2$/s and –560 m$^2$/s, respectively. Reductions during daytime are

also generally greater in inland regions than in coastal regions as shown is Figure 5d and 5e. Daytime

reductions in ventilation occur due to the combined effect of weakened wind speeds due to higher

surface roughness and changes (mostly decreases) in PBL heights (Figure S7). Changes in PBL heights

during daytime are mainly associated with air temperature changes because buoyancy production

dominants TKE during the day. Where there are larger air temperature decreases (increases), there is

reduced (increased) buoyancy production of TKE, which results in shallower (deeper) PBLs.

At night, spatially averaged ventilation coefficient increases by +8.2% (+24.3 m$^2$/s). As shown in

Figure 5f, greater ventilation growth occurs in coastal Los Angeles and Orange County, likely due to

higher PBL height increases (i.e., stemming from higher surface roughness increases from urbanization).

By contrast, in Riverside County, the effect of reductions in wind speed surpasses slight increases in

PBL heights, leading to reductions in atmospheric ventilation (Figure S7).

## 3.3 Effects of Urbanization on NOx and O$_3$ Concentrations due to Meteorological Changes

Concentrations of pollutants are profoundly impacted by meteorological conditions including air

temperature and the ventilation capability of the atmosphere (Aw and Kleeman, 2003; Rao et al., 2003).

This section discusses how meteorological changes due to land surface changes via urbanization in

Southern California affect gaseous pollutant concentrations (i.e., NOx and O$_3$).

### 3.3.1 Temporal and spatial patterns of NOx concentration changes and process drivers

As shown in Figure 6a, changes in meteorological fields due to urbanization have led to increases

in hourly NOx concentrations during the day (7 PST to 18 PST) and decreases at all other times of day. Peak increases in NOx of +2.7 ppb occur at 10 PST (i.e., for spatial mean values), while peak decreases of –4.7 ppb occur at 21 PST. Spatial mean changes in NOx concentrations are +2.1 ppb and +1.2 ppb in the morning and afternoon, respectively, and –2.8 ppb at night. In addition, daily 1-hour maximum NOx concentrations change only slightly: from 17.8 ppb at 6 PST in the Nonurban scenario to 17.9 ppb at 7 PST in the Present-day scenario.

Figures 7a,b,c show the spatial patterns of NOx concentration changes due to urbanization. In the morning (afternoon), most urban regions show increases in NOx concentrations (Figure 7a, b), with larger NOx concentration increases of up to +13.8 ppb (+5.5 ppb) occurring in inland regions compared to coastal regions. By contrast, NOx concentrations decrease at night across the region, with the largest decreases reaching –20.8 ppb. In general, greater decreases are shown in inland regions compared to coastal regions.

The spatial patterns of changes in NOx concentrations are similar to those for CO concentrations (Figure 7d,e,f). CO is an inert species and can be used as a tracer for determining the effect of ventilation on air pollutant dispersion since it includes accumulation effects of ventilation changes both spatially and temporally. Thus, the similarity in changes to NOx and CO spatial patterns suggests that NOx changes are driven by ventilation changes.     For example, at night, Riverside County shows decreases of up to –20.8 ppb in NOx concentrations (with corresponding decreases in CO of –119 ppb) despite suppressed ventilation at this location because of accumulative effects from coastal to inland regions.

### 3.3.2 Temporal and spatial patterns of O$_3$ concentration changes

As indicated by Figure 6b, O$_3$ concentrations in the lowest atmospheric layer decrease from 7 PST



to 11 PST, and increase during other times of day. The largest decrease of –0.94 ppb occurs at 10 PST, while the largest increase of +5.6 ppb occurs at 19 PST. Spatially averaged hourly $O_3$ concentrations

undergo a –0.6 ppb decrease, +1.7 ppb increase, and +2.1 ppb increase in the morning, afternoon, and night, respectively. Additionally, daily 1-hour maximum $O_3$ concentrations, which occurs at 14 PST in both scenarios, increases by +3.4%, from 41.3 ppb in the Nonurban scenario to 42.7 ppb in the Present-day scenario. The daily 8-hour maximum $O_3$ concentration increases from 38.0 ppb to 39.3 ppb (averaged over 11 PST to 19 PST in both scenarios).

Figure 7g,h,i show the spatial patterns of surface $O_3$ concentration changes. In the morning (Figure 7g), most regions show reductions in $O_3$ concentrations, with the largest decrease of up to –5.4 ppb in the inland regions. By contrast, most urban regions show increases in $O_3$ concentrations during the afternoon (Figure 7h), with the largest increase of +5.7 ppb occurring in Riverside County. Increases in $O_3$ concentrations are larger during night than the afternoon (Figure 7i), especially in the Riverside

County, with the largest increase in $O_3$ concentrations reaching +12.8 ppb.

### 3.3.3 Processes driving daytime and nighttime changes in $O_3$

The temporal and spatial patterns of changes in $O_3$ concentrations during the day suggest that these changes are mainly driven by the competition between (a) decreases in ventilation, which would tend to cause increases in $O_3$, and (b) the nonlinear response of $O_3$ to NOx changes. In the VOC-limited regime,

increases in NOx tend to decrease $O_3$ concentrations, and vice versa. (This explains why decreases in NOx emissions over weekends can cause increases in $O_3$ concentrations, a phenomenon termed the "weekend effect" (Marr & Harley, 2002).) The underlying cause of the weekend effect has to do with titration of $O_3$ by NO, as shown in R1.

$$NO + O_3 \rightarrow NO_2 + O_2 \hspace{4cm} (R1)$$



When NOx is high relative to VOC, R1 dominates NO to NO$_2$ conversion, which involves consuming O$_3$. In addition, increases in NO$_2$ can reduce OH lifetime due to increased rates of the OH + NO$_2$ reaction (R2), which is chain terminating.

$$NO_2 + OH + M \rightarrow HNO_3 + M \tag{R2}$$

In addition to these two aforementioned processes, changes in air temperature can also affect the

production rate of O$_3$, with higher temperatures generally leading to higher O$_3$ (Steiner et al., 2010).

In the morning when ventilation is relatively weak (shallow PBL and weak sea breeze), changes in NOx concentrations play an important role in driving surface O$_3$ concentrations. Regions with greater increases in NOx concentrations in general show greater decreases in O$_3$ concentrations (Figure 7g). Decreases in air temperature would also contribute to decreases in O$_3$ concentrations due to reductions

in O$_3$ production rates. In the afternoon when ventilation is strengthened (deep PBL, and stronger sea breeze), changes in both NOx concentrations and ventilation play important roles in determining O$_3$ concentrations (Figure 7h). Regions with higher increases in NOx concentrations tend to have lower increases in O$_3$ concentrations; this indicates that NOx increases (that would tend to decrease O$_3$) are counteracting decreases in ventilation (that would tend to increase O$_3$). In regions with relatively lower

increases in NOx concentrations and greater decreases in ventilation, such as Riverside County, increases in O$_3$ concentrations are larger.

At night, changes in O$_3$ concentrations are dominated by its titration by NO$_2$ as shown in (R3).

$$NO_2 + O_3 \rightarrow NO_3 + O_2 \tag{R3}$$

Where there are larger decreases in NOx concentrations (Figure 7c), there are greater increases in O$_3$

concentrations (Figure 7i), regardless of the magnitude of increases in atmospheric dilution (Figure 5f).



## 3.4 Effects of Urbanization on Total and Speciated PM$_{2.5}$ Concentrations due to Meteorological Changes

In this section, we discuss changes in total and speciated PM$_{2.5}$ mass concentrations due to urbanization. Total mass concentrations reported here only consider PM$_{2.5}$ generated from anthropogenic and biogenic sources mentioned in section 2.3, and exclude sea salt and dust. Speciated PM$_{2.5}$ is classified into three categories: (secondary) inorganic aerosols including nitrate (NO$_3^-$), sulfate (SO$_4^{2-}$) and ammonium (NH$_4^+$); primary carbonaceous aerosols including elemental carbon (EC), and primary organic carbon (POC); and secondary organic aerosol (SOA) including SOA formed from anthropogenic VOC precursors (ASOA) and biogenic VOC precursors (BSOA).

### 3.4.1 Temporal patterns of total and speciated PM$_{2.5}$ concentration changes

Figure 8 illustrates diurnal changes in total and speciated PM$_{2.5}$ concentrations due to meteorological changes attributable to urbanization. As suggested by Figure 8a, urbanization is simulated to cause slight spatially averaged increases in total PM$_{2.5}$ concentrations from 9 PST to 16 PST (up to +0.62 μg/m$^3$ occurring at 12 PST), and decreases during other times of day (up to –3.1 μg/m$^3$ at 0 PST). Increases in total PM$_{2.5}$ during 9 PST to 16 PST come from increases in primary carbonaceous aerosols, and nitrate; these species show hourly averaged concentration increases of up to +0.21, +0.14 μg/m$^3$, respectively. By contrast, BSOA decreases slightly during these hours. During other times of day, concentrations of all PM$_{2.5}$ species decrease dramatically. Inorganic aerosols, primary carbonaceous aerosols, and SOA show decreases of up to –1.7, –0.5 and –0.3 μg/m$^3$, respectively.

During morning hours, averaged hourly total PM$_{2.5}$ concentrations decrease by –0.20 μg/m$^3$, with 74% and 73% of the decrease contributed by changes in inorganic aerosols and SOA, respectively

(Figure 8). In the afternoon, spatially averaged total PM$_{2.5}$ concentrations increase by +0.24 μg/m$^3$. Primary carbonaceous aerosols contribute to half of the increase (+0.12 μg/m$^3$). For nighttime, total

PM$_{2.5}$ concentrations undergo a decrease of –2.5 μg/m$^3$, with 54% of the decrease attributed to changes in inorganic aerosols and 17% by primary carbonaceous aerosols.

### 3.4.2 Spatial patterns of total and speciated PM$_{2.5}$ concentration changes

Figure 9 presents spatial patterns of changes in total and speciated PM$_{2.5}$ due to urbanization. Decreases in concentrations prevail in urban regions during morning and night, whereas increases in
concentrations are dominant during the afternoon.

In the morning, the spatial pattern of changes in total PM$_{2.5}$ concentrations (Figure 9a) is similar to that of inorganic aerosols (Figure 9d). The west coastal region shows increases of up to +0.59 μg/m$^3$ in total PM$_{2.5}$ and +0.44 μg/m$^3$ in inorganic aerosols. In addition, increases occurring in some inland regions are caused by changes in primary carbonaceous aerosols. Other inland regions show decreases
of up to –4.2 μg/m$^3$ in inorganic aerosol and –2.7 μg/m$^3$ in total PM$_{2.5}$ concentrations.

In the afternoon, increases in total PM$_{2.5}$ (up to +1.4 μg/m$^3$, Figure 9b) include contributions from 1) inorganic aerosols (up to +0.8 μg/m$^3$, Figure 9e), 2) primary carbonaceous aerosols (up to +0.5 μg/m$^3$, Figure 9h), and 3) SOA (up to +0.3 μg/m$^3$, Figure 9.k).

At night, most regions within the Los Angeles metropolitan area show decreases in total PM$_{2.5}$ of –
3.0 to –6.0 μg/m$^3$ (Figure 9c) with contributions from all three categories of speciated PM$_{2.5}$.

### 3.4.3 Processes driving daytime and nighttime changes in PM$_{2.5}$

During the day, changes in speciated PM$_{2.5}$ concentrations are dictated by the relative importance of various competing pathways, including (a) reductions in ventilation causing increases in PM$_{2.5}$, (b)

changes in gas-particle phase partitioning causing increases (decreases) in $PM_{2.5}$ from decreases

(increases) in temperature, and (c) increases (decreases) in atmospheric oxidation from increases

(decreases) in temperature. Changes in ventilation appear to dominate the changes in primary

carbonaceous aerosols, as indicated by the similarity in spatial pattern to changes in CO, which can be

considered a conservative tracer (Figure 7d and 7e). As for semi-volatile compounds such as nitrate

aerosols (red dotted curve in Figure 8b) and some SOA species, concentrations increase during daytime

hours. This is because both decreased ventilation and gas-particle phase partitioning effects favoring the

particle phase (from temperature decreases) outweigh reductions in atmospheric oxidation.

Concentrations of sulfate and ammonium slightly increase due to urbanization (blue dotted curve in

Figure 8b). Since sulfate is nonvolatile, gas-particle phase partitioning does not affect sulfate

concentrations; lowered atmospheric oxidation rates due to reduced temperatures (which would tend to

decrease sulfate) nearly offset the effect of weakened ventilation (which would tend to increase sulfate).

In addition, BSOA concentrations are simulated to decrease (blue dotted curve in Figure 8d) due to

reduced biogenic VOC emissions, which occur due to reductions in both vegetation coverage and air

temperature from urbanization.

At night, decreases in $PM_{2.5}$ across urban regions are due to (1) enhanced ventilation owing to

deeper PBLs (relevant for all PM species), and (2) gas-particle phase partitioning effects that favor the

gas phase for semi-volatile compounds (i.e., nitrate aerosols and some SOA species) because of higher

air temperatures.

## 4. Conclusion

In this study, we have characterized the impact of land surface changes via urbanization on regional

meteorology and air quality in Southern California using an enhanced version of WRF/Chem-UCM.



The two main simulations of focus in this study are the "Present-day" and the "Nonurban" scenarios; the former assumes current land cover distributions and irrigation of vegetative areas, while the latter assumes land cover distributions prior to widespread urbanization and no irrigation. We assume identical anthropogenic emissions in these two simulations to allow for focusing on the effects of land

cover change on air pollutant concentrations.

Our results indicate that land surface modifications from historical urbanization have had a profound influence on regional meteorology. Urbanization has led to daytime reductions in air temperature for the lowest model layer and reductions in ventilation within urban areas. The impact of urbanization at nighttime shows the opposite effect, with air temperatures and ventilation coefficients

increasing. Spatially averaged reductions in air temperature and ventilation during the day are –0.6 K and –650 $m^2$/s respectively, whereas increases at night are +1.1 K and 24.3 $m^2$/s respectively. Changes in meteorology are spatially heterogeneous; greater changes are simulated in inland regions for (a) air temperatures decreases during day and increases during night, and (b) ventilation reductions during daytime. Ventilation at night shows increases in coastal areas and decreases in inland areas. Changes in

meteorology are mainly attributable to (a) increased surface roughness from buildings, (b) higher evaporative fluxes from irrigation, and (c) higher thermal inertia from building materials and increased soil moisture (from irrigation).

Changes in regional meteorology in turn affect concentrations of gaseous and particulate pollutants. NOx concentrations in the lowest model layer increase by +1.6 ppb during the day, and decrease by –

2.8 ppb at night, due to changes in atmospheric ventilation. $O_3$ concentrations decrease by –0.6 ppb in the morning, and increase by +1.7 (2.2) ppb in the afternoon (night). Decreases in the morning and increases during other times of day are more noticeable in inland regions. Changes in $O_3$ concentrations are mainly attributable to the competition between (a) changes in atmospheric ventilation, and (b)

changes in NOx concentrations that alter $O_3$ titration. Note that while changes in air temperature can also influence $O_3$ concentrations during the day, this effect is overwhelmed by changes in ventilation and concentrations of NOx in our study. As for $PM_{2.5}$, total mass concentrations increase by +0.24 μg/m$^3$ in the afternoon, and decrease by –0.20 (–2.5) μg/m$^3$ in the morning (night). The major driving processes of changes in $PM_{2.5}$ concentrations are (a) changes in atmospheric ventilation, (b) changes in gas-particle phase partitioning for semi-volatile compounds due to air temperature changes, and (c) 565 changes in atmospheric chemical reaction rates from air temperature changes.

Our findings suggest that increases in evapotranspiration, thermal inertia, and surface roughness due to historical urbanization are the main drivers of regional meteorology and air quality changes in Southern California. During the day, urbanization has led to regional air temperature reductions but increased ozone and $PM_{2.5}$ concentrations. During nighttime, urbanization has led to increases in 570 regional air temperatures and $O_3$ concentrations, but decreases in NOx and $PM_{2.5}$ concentrations. Our study provides insight into the potential impacts of land surface modifications via urbanization on regional meteorology and air quality, and can be informative for decision making on sustainable urban planning to achieve a balance between climate mitigation/adaptation and air quality improvements.

## Author Contributions

GBW designed the study. YL performed the model simulations, carried out data analysis, and wrote the manuscript. GBW and DS mentored YL. JZ contributed to the setup of WRF/Chem-UCM. All authors contributed to editing the paper.



## Competing Interests

The authors declare that they have no conflict of interest.

## Acknowledgements

This research is supported by the US National Science Foundation under grant CBET-1512429, grant CBET-1623948 and grant CBET-1752522. Model simulations for the work described in this paper are supported by the University of Southern California's Center for High-Performance Computing (https://hpcc.usc.edu/). We thank Scott Epstein and Sang-Mi Lee at South Coast Air Quality

Management District, Jeremy Avise at California Air Resources Board for providing us emission datasets. We also thank Ravan Ahmadov and Stu McKeen at National Oceanic and Atmospheric Administration for their helpful suggestions.

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





# Figures and Tables

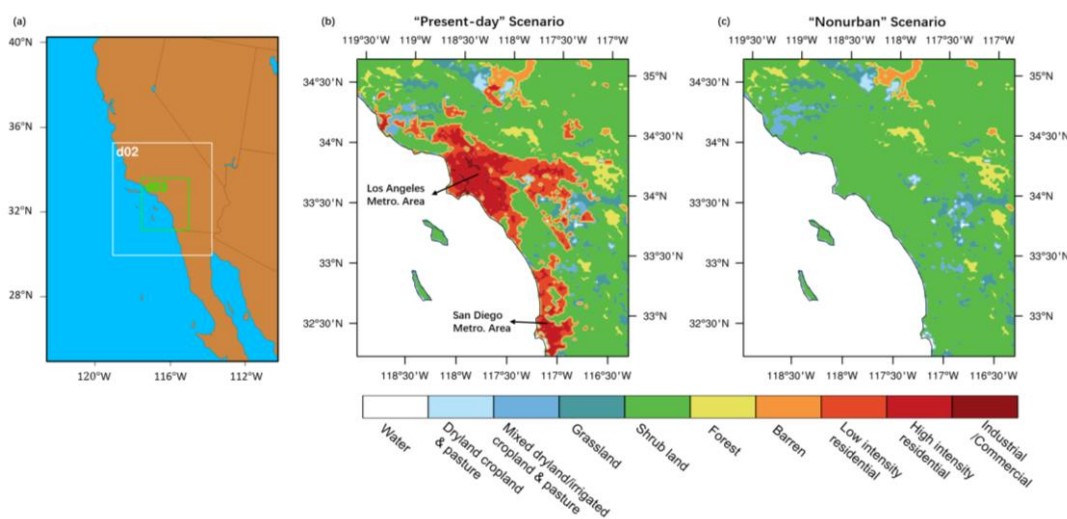

**Figure 1.** Maps of (a) the three nested WRF/Chem-UCM domains, and (b,c) land cover types for the innermost
domain (d03) for the (b) Present-day and (c) Nonurban scenarios.



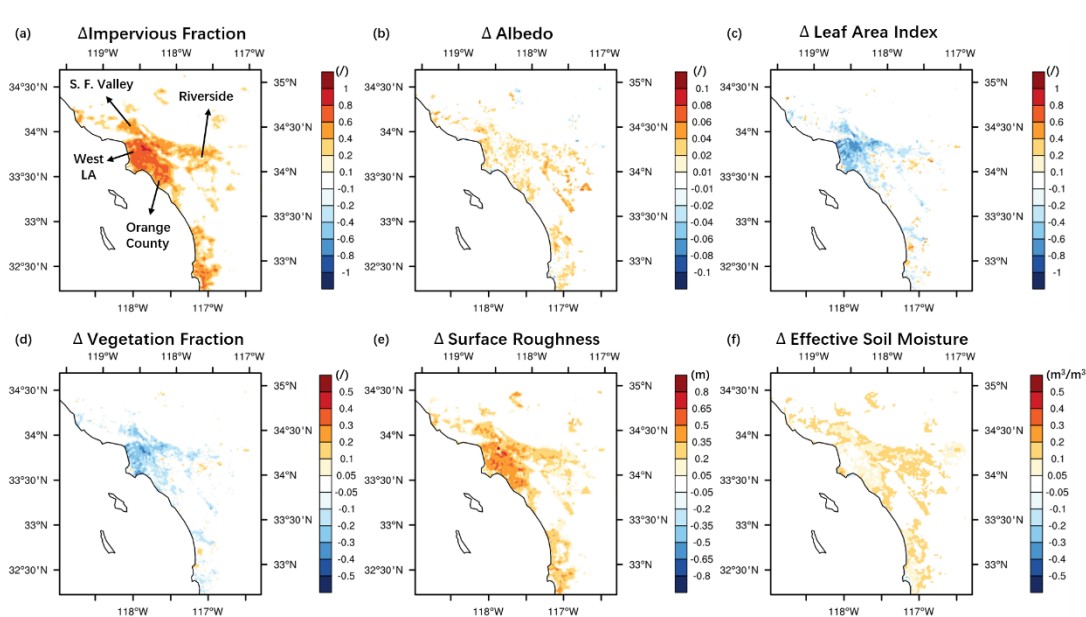

**Figure 2.** Spatial patterns of differences (Present-day – Nonurban) in land surface properties for urban grid cells.
Panels (a) to (f) are changes in impervious fraction, albedo, leaf area index (LAI), vegetation fraction (VEGFRA),
surface roughness, and effective soil moisture, respectively. Effective soil moisture is calculated as the product of
pervious fraction for urban grid cells (1 – impervious fraction) and soil moisture for the pervious portion of the grid
cell.





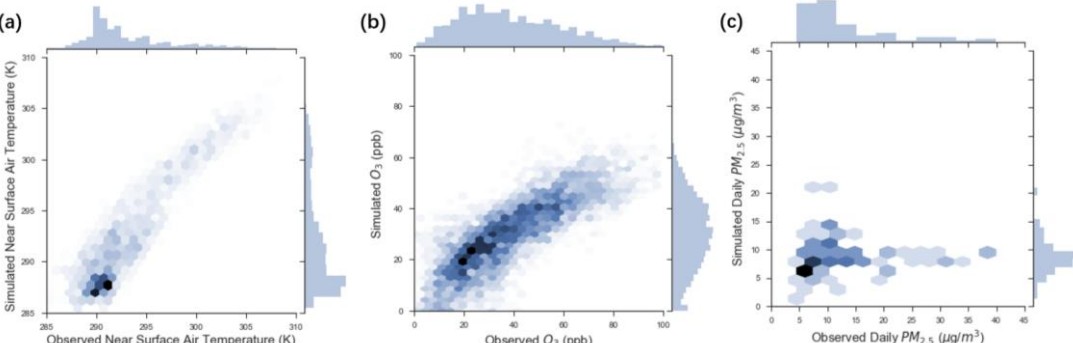


**Figure 3.** Comparison between modeled and observed (a) hourly near-surface air temperature (K), (b) hourly $O_3$ concentrations (ppb), and (c) daily $PM_{2.5}$ concentrations ($\mu g/m^3$). Note that daily $PM_{2.5}$ concentrations from simulations include sea salt, but exclude dust. Darker hexagonal bins correspond to higher point densities in the scatter plots. Histograms of both observations and modeled values are also shown at the edges of each panel.


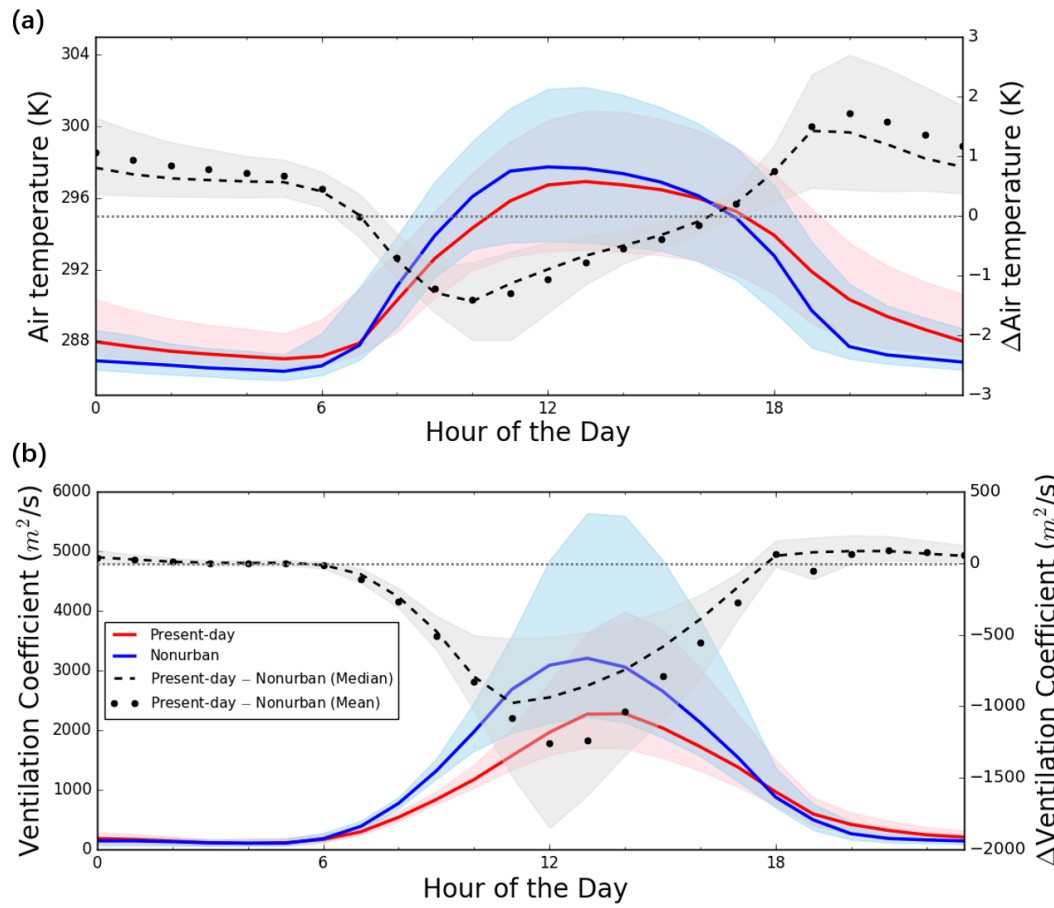

**Figure 4.** Diurnal cycles for present-day (red), nonurban (blue), and present-day – nonurban (black) for (a) air temperature in the lowest atmospheric layer (K) and (b) ventilation coefficient (m²/s). Values are obtained by averaging over urban grid cells and the entire simulation period for each hour of day. The solid and dashed curves give the median values, while the shaded bands show 25th and 75th percentiles. Dots indicate mean values for differences between Present-day and nonurban. The horizontal dotted line in light grey shows $\Delta = 0$ as an indicator of positive or negative change by land surface changes via urbanization.




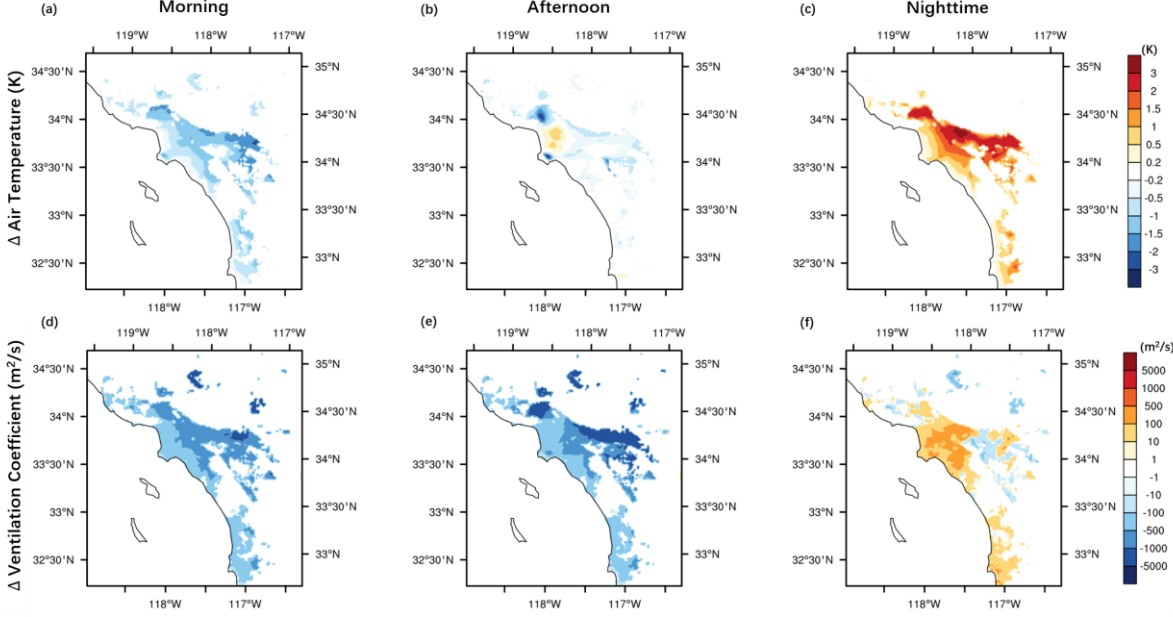

**Figure 5.** Spatial patterns of differences (Present-day – nonurban) in temporally averaged values during morning, afternoon and nighttime for (a,b,c) air temperature in the lowest atmospheric layer, and (d,e,f) ventilation coefficient. Morning is defined as 7 PST to 12 PST, afternoon as 12 PST to 19 PST, and nighttime as 19 PST to 7 PST. We refer to morning and afternoon as daytime. Note that values are shown only for urban grid cells.

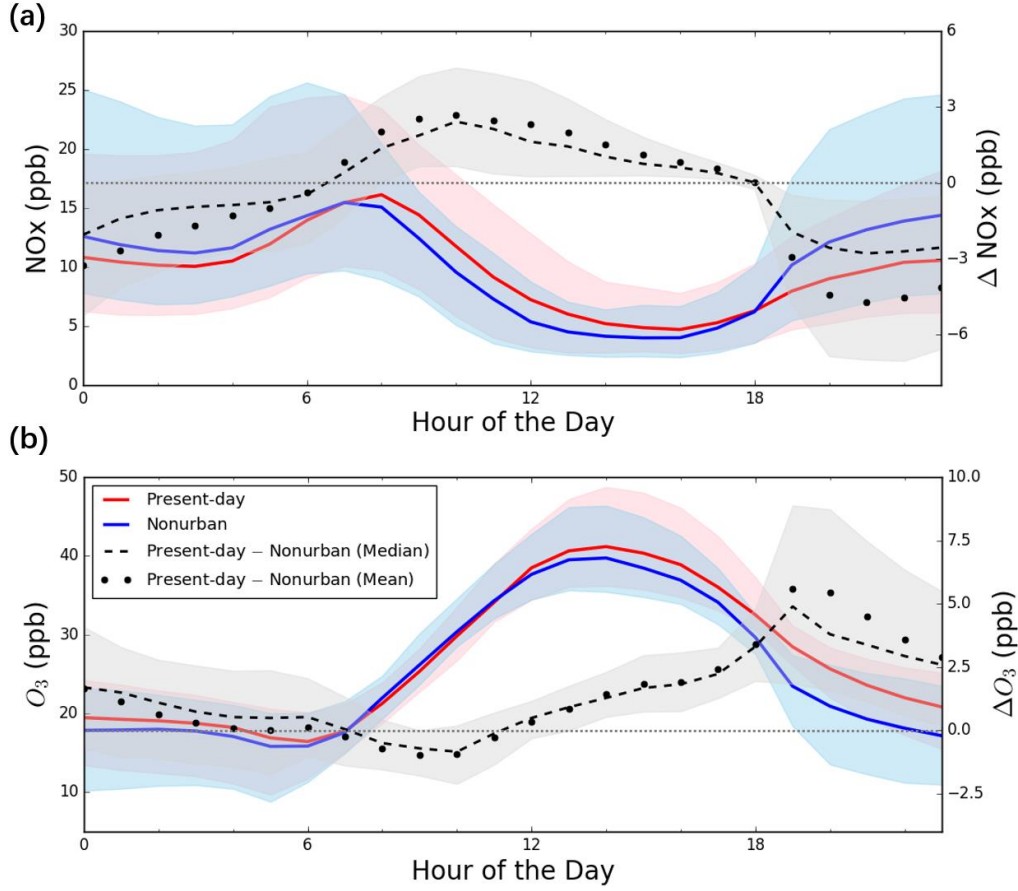


**Figure 6.** Diurnal cycles for present-day (red), nonurban (blue), and present-day – nonurban (black) for (a) NOx (ppb) and (b) $O_3$ concentrations (ppb). Values are obtained by averaging over urban grid cells and the entire simulation period for each hour of day. The solid and dashed curves give the median values, while the shaded bands show 25th and 75th percentiles. Dots indicate mean values for differences between Present-day and nonurban. The horizontal dotted

line in light gray shows $\Delta = 0$ as an indicator of positive or negative change by land surface changes via urbanization.





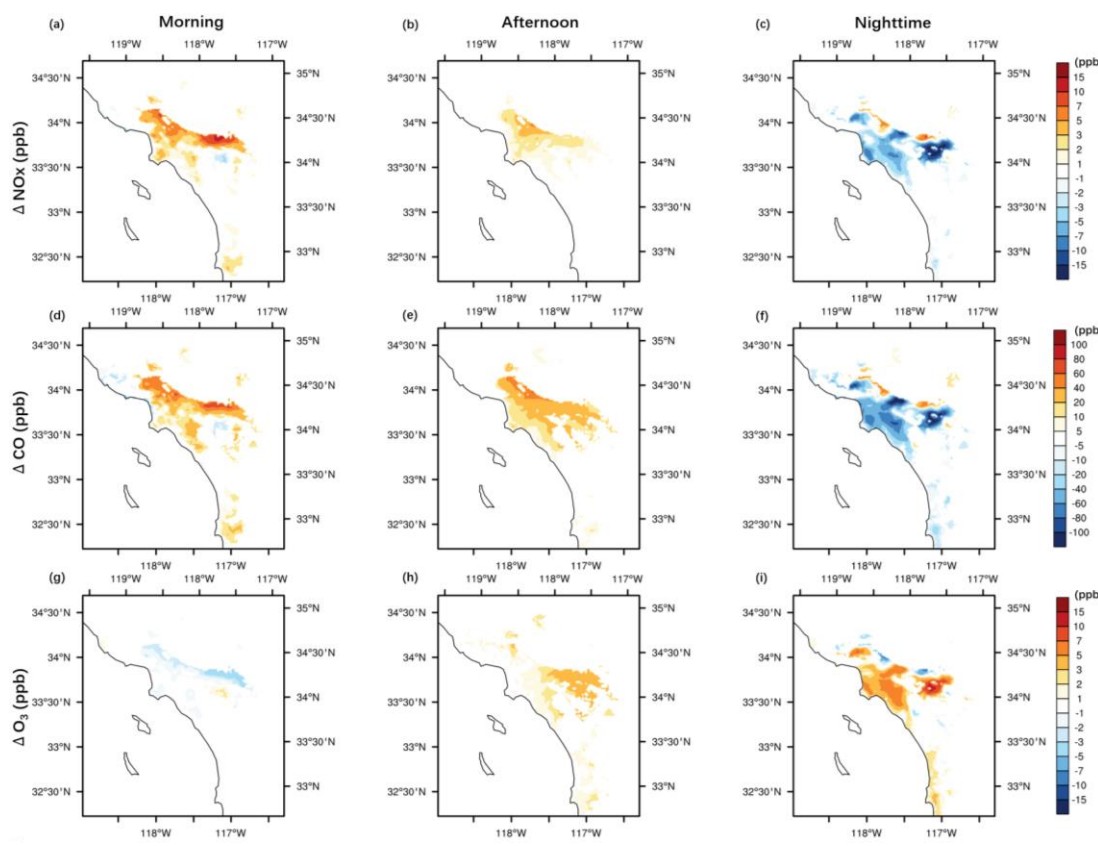

**Figure 7.** Spatial patterns in differences (Present-day – nonurban) of temporally averaged values during morning, afternoon and nighttime for (a,b,c) NOx, (d,e,f) CO, and (g,h,i) O$_3$ concentrations. Morning is defined as 7 PST to 12 PST, afternoon as 12 PST to 19 PST, and nighttime as 19 PST to 7 PST.





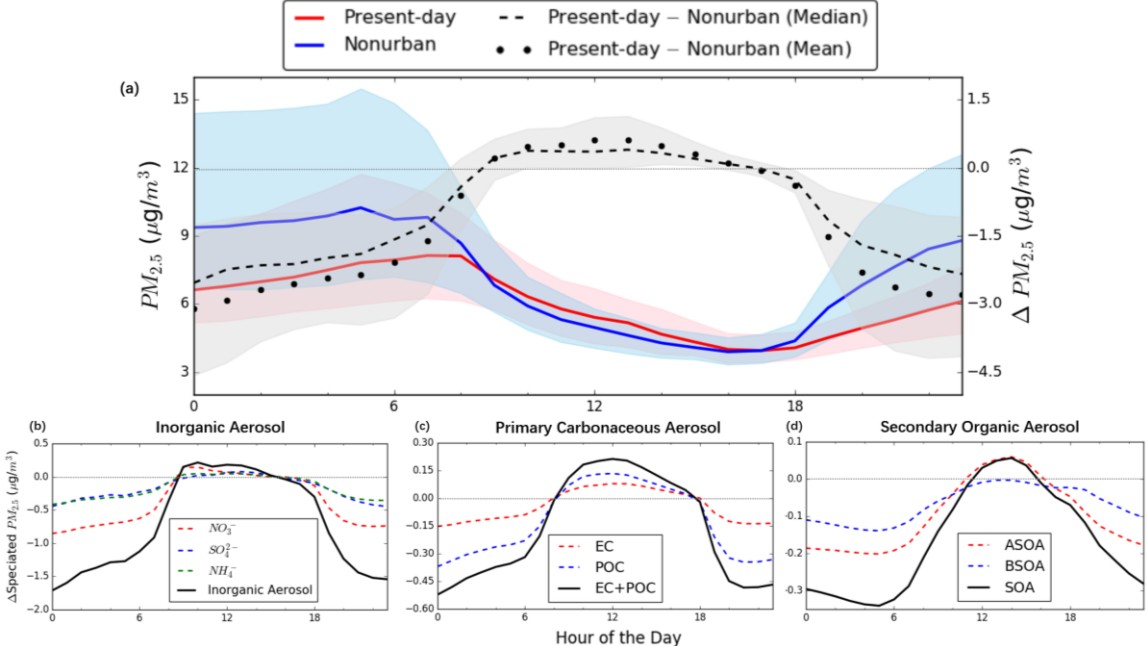

**Figure 8.** Diurnal cycles for spatially averaged PM$_{2.5}$ concentrations. Panel (a) shows Present-day, nonurban, and present-day – nonurban for total PM$_{2.5}$ (excluding sea salt and dust). The lower row shows differences (Present-day – nonurban) in speciated PM$_{2.5}$ including (b) inorganic aerosols (NO$_3^-$, SO$_4^{2-}$, NH$_4^+$), (c) primary carbonaceous aerosols (EC, POC), and (d) secondary organic aerosols (ASOA, BSOA). The horizontal dotted line in light grey is shown for $\Delta = 0$ as an indicator of positive or negative change by urbanization.





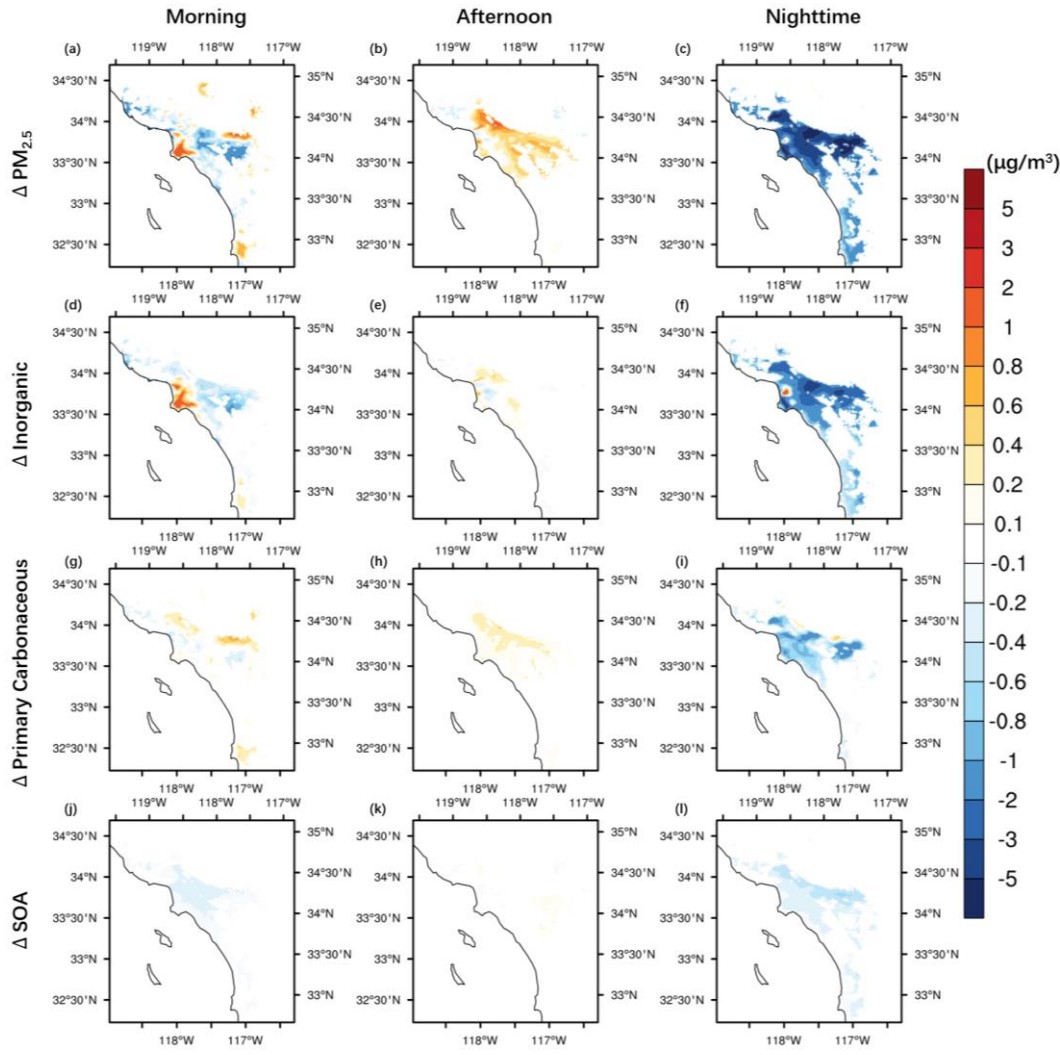

**Figure 9.** Spatial patterns in differences (Present-day – nonurban) of temporally averaged values during morning, afternoon, and nighttime for PM$_{2.5}$. Panels (a)–(c) show total PM$_{2.5}$; (d)–(f) inorganic aerosol; (g)–(i) primary carbonaceous aerosol; and (j)–(l) secondary organic aerosol. Morning is defined as 7 PST to 12 PST, afternoon as 12 PST to 19 PST, and nighttime as 19 PST to 7 PST.





**Table 1.** Summary statistics (mean bias (MB), normalized mean bias (NMB), mean error (ME), and root mean square error (RMSE)) for model evaluation, which compares simulated hourly near-surface air temperature (T2), hourly $O_3$ and daily $PM_{2.5}$ concentrations to observations.

| Variable | N [a] | Mean | | MB [b] | NMB [c] | ME [d] | RMSE [e] |
| --- | --- | --- | --- | --- | --- | --- | --- |
| | | Observations | Simulations | | | | |
| T2 | 1944 | 293.0 K | 292.0 K | -1.0 K | -0.3% | 1.9 K | 2.2 K |
| $O_3$ | 5171 | 38.7 ppb | 30.0 ppb | -8.7 ppb | -22% | 11.8 ppb | 14.6 ppb |
| $PM_{2.5}$ | 81 | 12.9 $\mu g/m^3$ | 9.2 $\mu g/m^3$ | -4.0 $\mu g/m^3$ | -31% | 6.2 $\mu g/m^3$ | 9.5 $\mu g/m^3$ |

[a.] Total number of data points comparing modeled versus observed values across all measurement station locations over the simulation period

[b.] $MB = \frac{1}{N}\sum(mol_i - obs_i)$

[c.] $NMB = \frac{\sum(mol_i - obs_i)}{\sum obs_i}$

[d.] $ME = \frac{1}{N}\sum|mol_i - obs_i|$

[e.] $RMSE = [\frac{1}{N}\sum(mol_i - obs_i)^2]^{\frac{1}{2}}$