# Peer review of "Effects of Urbanization on Regional Meteorology and Air Quality in Southern California"

_Atmospheric Chemistry and Physics, 2018_

## Referee Comment (RC1) · Anonymous Referee #1 · 19 Oct 2018

The manuscript presents two sets of simulations realized with the model WRF-CHEM coupled with the Single Layer Urban Canopy Model, over the Los Angeles region for a 10 days period at the end of June-beginning of July 2012. One set of simulations is realized with the current landuse, including the urban area of Los Angeles. The second set is realized replacing the urban area with shrub, representing the original vegetation (as claimed by the authors). The anthropogenic emissions are the same for both simulations. By comparing the results of the two simulations, authors derive the impact of urbanization on meteorology and air quality in the region.

I have two main comments to this manuscript.

a) Authors rely heavily on previous work by the same team (mainly by Vahmani) to justify the set-up used, and the improvements obtained in simulating air temperature (for

example due to the inclusion of the irrigation system). However, at lines 358-361, they say that all the previous simulations were performed without accounting for the shadowing effect in the street canyon, and with a different technique to estimate the surface temperature. On the contrary, the simulations presented in the manuscript consider shadowing and use the default formulation to estimate the surface temperature for impervious surfaces. The impact on the results of these different modeling choices seems important to the point that with the new approach urbanization decreases daytime temperature compared to the non-urban case, while with the previous set-up urbanization increased the daytime temperature. While I certainly agree that it is important to account for shadowing, I think that it is necessary to perform a more thorough validation of the simulations to get more confidence in the results, also because the RMSE, presented in table 1, is much larger than the urbanization effect. Therefore, I recommend making a separate analysis of urban and rural stations, and to separate between urban stations based on the different urban morphological characteristics. The validity of this study relies completely on the model capability to reproduce correctly the differences between urban and rural areas, so it is very important to show this comparison. For example, the following questions should be addressed: what are the RMSE and Mean Bias for the urban stations only? And for the rural stations? We have to be sure that the model is simulating correctly the urban areas AND the rural areas (in particular shrubs). Is the model able to capture the maximum and minimum temperature at each station? Is the model able to reproduce the differences between stations, and in particular the differences between the urban and the rural stations? (e. g. if at a certain hour higher temperature is measured in an urban station compared to a rural one, is the model doing the same? If rural stations measured lower minimum (maximum) than urban stations, is the model doing the same qualitatively and quantitatively?, etc.).

b) It must be made clear that the simulation with current anthropogenic emissions, but not the city, is a hypothetical one – there cannot be emissions without a city. In the last sentence of the manuscript (lines 570-574), authors say that their results "can be informative for decision making on sustainable urban planning to achieve a balance

between climate mitigation/adaptation and air quality improvements". Honestly, I do not see how. This type of studies may have a scientific value, in the sense that they demonstrate the importance of taking into account the presence of the city in the simulation of air quality and meteorology (it would be interesting to see if the simulation with the city provides better results compared to measurements than the simulation without the city). But I do not see how they can be helpful for urban planning. Replacing the city with shrubs cannot certainly be considered a strategy to manage urban climate or improve air quality. The differences that authors estimated between the urban and the no-urban simulations are not the maximum difference that can be obtained managing the landuse. They actually do not give any information about the impact of any realistic mitigation strategy based on landuse management. I think it is very important that authors clarify what they have in mind because this is at the basis of the motivation of the whole manuscript.

Detailed comments:

1) Lines 64-66. Urban regions in semi-arid or arid surroundings have a weak (or non-existent) daytime UHI, but they have a very strong nocturnal UHI. I think authors missed the fundamental difference between daytime and nighttime UHI, (being the latter the most frequent).

2) Line 168. On which basis authors claim that the period chosen is representative of summer conditions in Southern California?

3) Line 174. Please provide the value of the depth of the lowest model level.

4) Line 215. Is the irrigation module implemented just for the pervious fraction of the urban cells, or also for the rural cells (to account for agricultural crops in the region)?

5) Line 302. I would avoid indicating the percentage for temperature. This would depend on the unit (if you use Celsius or Kelvin). I would just put degrees.

6) Line 303. On which basis authors claim that this is "acceptable".

[Figure]

7) Section 3.2.3. I suggest studying the difference in sea breeze front progression between the two cases (urban and no-urban). This will give a better understanding of what is happening.

8) Lines 335-338. This is not clear. Before it is said that urbanization decreases temperatures and not increases.

9) Line 370. During night time atmosphere cools. The energy stored in the building during daytime (what authors call upward ground heat flux, I suppose) reduces the cooling. The higher PBL in the urban simulation will reduce the cooling too because the effect of the surface cooling is distributed in a greater depth than in the no-urban case. The two mechanisms (energy stored in buildings, and high PBL), both reduce cooling. They do not compete they go in the same direction.

10) Lines 375-376. Same as above, during the night there is not heating, there is cooling.
* * *

---

## Referee Comment (RC2) · Anonymous Referee #2 · 5 Dec 2018

Dear authors, The paper is well written and clearly structured, however I would recommend a number of major changes in order to be suitable for publication. Please find my comments below:

General: I see a general problem in the definition of the scope of the study. A 'before human settlement' scenario should not consider emissions at all and further describes a period about ∼100-150 years ago which means that you would also have to consider a different climate period, land use etc.. I definitely would recommend to re-define the scope of the study, because in the current state, just distinguishing between 100% urban vs. 0% urban is not sufficient to analyze the above mentioned scenario.

I am further not fully convinced about the added benefit of this study for sustainable urban planning recommendations. I am aware that these model systems are not suitable

for applied urban planning, but however the currently existing urban canopy models in WRF-Chem (and other models), together with high resolution datasets for both emission and urban morphology do offer a framework for a number of different scenarios in the context of climate change/UHI mitigation. Recent studies have been analyzing the impact of highly reflecting building materials, urban greening or varying building density for a number urban areas. These aspects should also be possible with this model system and worth being discussed in order to increase the scientific substance of that work and highlight the new contribution to the field. In light of the scope of the journal, it should also be worked out more detailed what are the implications for atmospheric science in general rather than purely investigating local/regional aspects.

I am convinced, that the model system, combined with the emission and land surface data sets offer a promising tool for discussing air quality/meteorology interactions in large urban areas such as Los Angeles, but however think that the variety of scenarios should be increased in order to allow for a more robust results towards currently relevant issues. The authors rely mostly on previous work with equal model configuration. Therefore, the own contribution to the field and the new development does not come out clearly. The paper however is well written and easy to follow, but crucial points have to be considered in a review before being able undertake a detailed line-by-line evaluation.

1. The scope of the study should be defined more clearly in light of the above mentioned points. The experimental design should be expanded, in order to include more own ideas/developments.

2. One interesting and highly relevant point in my opinion is the 'irrigation' module which might offer a nice tool for testing different irrigation scenarios.

3. Why did you select a single-layer urban canopy model rather that a more complex multi-layer canopy representation (BEP/BEM)? The latter should deliver higher accuracy close to the ground I guess? What is the depth of the lowest model level?

4. Where do the input parameters for SLUCM come from?

5. What is the additional gain of a 30 m land surface classification which has to be scaled to 2 km model resolution?

6. Is there a problem with regard to the discrepancy between emission inventory and model resolution?

7. How realistic is the surrounding 'non-urban' land use classification for the 'historical' scenario?

8. How well does the model simulate urban AND rural parameters?

9. Please specify how results from this study can serve as contribution for applied urban planning?

10. In relation to other chapters, the introduction is slightly too long. Try to focus on the relevant points here and shorten were possible.

Please find below comments for specific sections, which partly have been addressed in the main points above.

Ln 11: ventilation not a good expression here

13: 'before human settlement' is a bit misleading here, as it is not entirely captured by your model setup. As mentioned before, some effort has to be put in a clear definition of the scope of your study. What problem should be addressed – also in light of recommendations for real urban planning (Lines 570-573?

43: "Differences in surface temperature..." What was the purpose of these studies mentioned here and what do they try to answer? How does this sentence relate to your study and the intention for this work?

47: "UCI": How does this relate to your study?

67: What is the role of the atmospheric aerosol burden for UHI formation?

[Figure]

73: better "characteristics/shape of the PBL is dependent on..."

81: better "due to urbanization..."

86: unclear what is meant by "meteorological changes via altered emissions,..."

115: Why do higher PBLs increase PM 2.5 concentration? Please discuss the related processes here.

119: How exactly does your experimental setup treat the "wide heterogeneity of urban land surface processes" compared to existing studies? A large number of studies already exist using model systems (e.g. WRF) which include urban canopy models with varying complexity (SLUCM, BEP), which consider a similar level of heterogeneity than your experiments? Please discuss your statement.

122: unclear expression "amongst"?

134-140: It should be made clear which new aspects you aim to analyze compared to the studies mentioned above. In my opinion simply turning urban on/off does not reveal significantly new insights. Further the term "human disturbance" is unclear, as this would also involve air quality modifications.

174: Please specify your lowest model level.

175: "process parametrization" unclear

180: please discuss the term "real world representation", answering the question why the WRF default land use classification in WRF is not "real" enough for your case comparing these datasets with your input. What was the idea behind using a 30m dataset? Please briefly discuss the gain of using 30 m land cover data for a maximum resolution of 2 km. How much information "is lost" by the process of "upscaling" the LU data. Would the 2011 NLCD dataset add additional benefit?

205: did you use the additional sub-tiling option in WRF?

243: Do you consider daily emission profiles? Meaning, do you find two "peaks" in for instance in NOx emission/concentration?

267: How realistic is the conversion to shrub-land for all grid cells? Would you expect different effects for a non-urban, but more heterogeneous "before human" land cover?

294: Please indicate better proof of the "good fit" mentioned here. It is not indicated by Figures S1 and 3 for pm2.5. How does the correlation coefficient look like? What are the reasons for the poor correlation especially for the high range of the observed concentrations? How representative are the measurement stations? As the ozone concentration is highly dependent on temperature you find a good fit. Does the poor fit for PM 2.5 relate to high mixing, chemistry, both? How do correlations look like for NO2,NO,CO? Are the simulated diurnal variations realistic? Please also discuss the values from Table 1? Are they particularly good/bad?

347: can you find impacts on the strength of the sea breeze when there is no urban area left?

363: what is the order of difference between shadow model on/off?

367: origin of the UCM parameters?

381: calculation of the ventilation coefficient?

386: please evaluate the quantity values here? Provide relative numbers

395: What is the relation between PBL height and surface roughness? Please provide more details. Can you find proof for this in your study?

490: can you say something about the change of PBL dynamic comparing urban and non-urban. I suspect concentration of PM 2.5 is highly dependent on the boundary layer depth. Expecting lower PBLs in "urban-free" areas actually should decrease PM 2.5 in summer?

530: What happens to the pbl height in non-urban environment? Even deeper?

535: specify "enhanced"

541: how confident are you that the land use class in the "before-human" settlement is correct? Or is it just a guess?

573: As mentioned earlier I am not entirely convinced, how findings from this study could be used for applied urban planning? You mention 'mitigation and adaptation', but a complete 'removal' of the urban area should be hard to transfer into an actual applicable strategies. Maybe more 'moderate' scenarios would be better. However, avoiding a complete re-doing of model experiments, the scope of the study should be formulated differently.
* * *

---

## Referee Comment (RC3) · Anonymous Referee #2 · 6 Dec 2018

Please find more information on a similar study for further reference: Zhong, S., Qian, Y., Zhao, C., Leung, R., Wang, H., Yang, B., ... & Liu, D. (2017). Urbanization-induced urban heat island and aerosol effects on climate extremes in the Yangtze River Delta region of China. Atmospheric Chemistry and Physics, 17(8), 5439-5457.

---

## Referee Comment (RC4) · Anonymous Referee #3 · 29 Dec 2018

This manuscript investigates impacts of urbanization in Southern California on regional meteorology and air quality. Simulations using an innermost domain with 2 km resolution are conducted by WRF-Chem coupled with UCM. The simulations are driven by current climate and anthropogenic emissions with and without urban pixels and are applied to characterize impacts of historical urbanization on regional and temporal distributions of temperature and concentrations of NOx, O3, and PM2.5. The authors conclude that urbanization causes daytime decreases in temperature and increases in O3 and PM2.5. In the nighttime, the simulation results present nighttime increases in temperature and O3, while the concentrations of NOx and PM2.5 show reductions. The authors attribute these changes to urban-induced modifications in various competing drivers including irrigation, thermal properties of building materials and surface

roughness.

General comments:

The topic addressed is interesting and relevant to ACP readers. However, I have reservations about the robustness of the conclusions presented. In my opinion, significant revisions with new analysis and more careful model verification of the simulations are required.

The impact of urbanization is derived from the differences between temperature and concentrations of fields simulated by a WRF-Chem configuration that includes urban pixels and by a scenario where urban pixels were converted to shrub. This methodology has been presented in previous work and the nighttime impact of urbanization has been well documented in the literature. For instance, the paper by Li et al ("Achieving accurate simulations of urban impacts on ozone at high resolution", ERL, 9, 2014) introduced similar configurations (WRF-Chem including anthropogenic emissions, with and without urbanization) and used them to derive impacts of urbanization on air quality by analyzing the differences in the simulated fields between the two scenarios. Although the region and the period of time considered in this manuscript are different, the main idea and the nighttime impact are similar. The daytime impact reported in this manuscript is questionable because its magnitude shows values smaller than the model error (see specific comment 2). Careful analysis of the robustness of the impact is needed, especially given that this impact conflicts with previous results as reported (Line 355). The authors need to emphasize what is new related to this research and how it advances the existing research on the topic.

Specific comments:

1-It is unclear why the authors chose a 10-day period of the summer of 2012? And in what basis the period chosen is "representative of typical summer days in Southern California"? Why not using more years?

2-The statistics presented in Table 1 indicate that the impact of urbanization is smaller than the model error for all the fields analyzed. For example, the magnitude of the simulated change in O3 is less than 5.6 ppb (Line 429). The mean and root mean square errors reported in Table 1 are 11.8 ppb and 14.6 ppb, respectively. Thus, the impact described, which is the main conclusion of the manuscript, is not robust given that it lies within the model error. Perhaps, simulations using other years could increase the statistical significance of the results presented.

3-The authors state in the conclusion that "...due to historical urbanization are the main drivers of regional meteorology and air quality changes in Southern California" (Line 567). However, the simulations presented in the manuscript cannot be applied to reach such conclusion. There are several critical factors that are not accounted for. For example, the initial and boundary conditions use current atmospheric conditions and therefore do not include the effect of climate change. The amount of the background $CO_2$ concentration specified in WRF is fixed (assuming that both configurations use the same setup except for urbanization as stated). The anthropogenic emissions did not exist before human settlement. I suggest that the authors rephrase their motivation and conclusion, and simply focus on the impact of urbanization without attributing historical changes solely to urbanization.

4-There are some claims that need clarification. For example the authors state in line 152 "In this study, we couple WRF/Chem to the urban canopy model (UCM)..." However, the WRF/Chem model is already coupled to UCM. I believe what the authors did is activating the option for this coupling. In line 180 "we update the default WRF/Chem to include a real-world representation of land surface physical properties and processes..." But again, the options for using NLCD and NUDAPT for land surface representations are available within WRF. Please clarify what is meant by "we update the default WRF/Chem".

5-The ability of WRF-Chem to realistically represent urban processes requires more evaluation to better establish the credibility of the present-day scenario. The comparison between observations and simulations shown in Fig. 3 does not indicate to me a "good fit at lower values" as stated in line 294. The observed low values of temperature are around 290 K, but the simulated temperature shows low values of 287K. The difference between these values is larger that the impact reported. Therefore, better model verification should be considered. I also suggest adding to Fig. 3 panels comparing diurnal variations of observed and simulated temperature, O3 and PM2.5 (similar to Fig. 4a).

6- Figs 5, 7 and 9 include values of simulated fields within urban grid cells only. The authors should consider superimposing in these figures values for the entire domain including nonurban grid cells. It would be very helpful to see the differences in the simulated fields within both urbanized pixels and also grid cells that remain natural in both scenarios considered.

---

## Author Comment (AC1) · 13 Feb 2019

**Response to Anonymous Referee #1**
(Note: Reviewer comments are listed in grey, and responses to reviewer comments are in black. Pasted text from the new version of the paper is in italics.)

The manuscript presents two sets of simulations realized with the model WRF-CHEM coupled with the Single Layer Urban Canopy Model, over the Los Angeles region for a 10 days period at the end of June-beginning of July 2012. One set of simulations is realized with the current land use, including the urban area of Los Angeles. The second set is realized replacing the urban area with shrub, representing the original vegetation (as claimed by the authors). The anthropogenic emissions are the same for both simulations. By comparing the results of the two simulations, authors derive the impact of urbanization on meteorology and air quality in the region.

We greatly appreciate the reviewer's helpful comments. We believe that addressing his/her comments have greatly improved the quality of our paper.

I have two main comments to this manuscript.
a)     Authors rely heavily on previous work by the same team (mainly by Vahmani) to justify the set-up used, and the improvements obtained in simulating air temperature (for example due to the inclusion of the irrigation system). However, at lines 358-361, they say that all the previous simulations were performed without accounting for the shadowing effect in the street canyon, and with a different technique to estimate the surface temperature. On the contrary, the simulations presented in the manuscript consider shadowing and use the default formulation to estimate the surface temperature for impervious surfaces. The impact on the results of these different modeling choices seems important to the point that with the new approach urbanization decreases daytime temperature compared to the non-urban case, while with the previous set-up urbanization increased the daytime temperature. While I certainly agree that it is important to account for shadowing, I think that it is necessary to perform a more thorough validation of the simulations to get more confidence in the results, also because the RMSE, presented in table 1, is much larger than the urbanization effect. Therefore, I recommend making a separate analysis of urban and rural stations, and to separate between urban stations based on the different urban morphological characteristics. The validity of this study relies completely on the model capability to reproduce correctly the differences between urban and rural areas, so it is very important to show this comparison. For example, the following questions should be addressed: what are the RMSE and Mean Bias for the urban stations only? And for the rural stations? We have to be sure that the model is simulating correctly the urban areas AND the rural areas (in particular shrubs). Is the model able to capture the maximum and minimum temperature at each station? Is the model able to reproduce the differences between stations, and in particular the differences between the urban and the rural stations? (e.g., if at a certain hour higher temperature is measured in an urban station compared to a rural one, is the model doing the same? If rural stations measured lower minimum (maximum) than urban stations, is the model doing the same qualitatively and quantitatively? etc.).

We thank the author for bringing up these important points. Our detailed response is listed below.

Firstly, after a careful comparison among different model set-ups, we found that the

parameterization of surface temperature is the more important factor that affects daytime air temperature compared to whether or not we account for shadow effects in urban canopies (i.e., see Figure S16 in the supplemental information). Therefore, we delete "and shading effects within urban canopies" in the sentence "In addition, increases in thermal inertia caused by use of manmade materials (e.g., pavements and buildings) and shading effects within urban canopies can contribute to simulated temperature reductions during the morning." We also modified the last paragraph in section 3.2.3 as below.

*"Note that changes in air temperature during daytime shown here disagree with Vahmani et al. (2016). While our study detects daytime temperature reductions due to urbanization, Vahmani et al. (2016) suggests daytime warming. After detailed comparison of the simulations in our study versus Vahmani et al. (2016), we find that the differences are mainly associated with UCM configuration. First, our study uses model default calculations of surface temperature for the impervious portion of urban grid cells, whereas Vahmani et al. applied the alternative calculation proposed by Li and Bou-Zeid, 2014. Li and Bou-Zeid, 2014 intended the alternate surface temperature calculation to be performed as a post-processing step rather than during runtime. After a careful comparison among different model set-ups, we found that the parameterization of surface temperature is an important factor that affects simulated daytime air temperature (See Figure S16). Second, our study accounts for shadow effects in urban canopies, whereas Vahmani et al. (2016) assumes no shadow effects. (We note here that the default version of the UCM has the shadow model turned off. The boolean SHADOW variables in module_sf_urban.F needs to be manually switched to true to enable the shadow model calculations. With the shadow model turned off, all shortwave radiation within the urban canopy is assumed diffuse.) We suggest that it is important to include the effects of building morphology on shadows within the canopy, and to track direct and diffuse radiation separately, and therefore perform simulations in this study with the shadow model on. Note that the effect of shadowing is not as significant as the parameterization of surface temperature in our study, because the ratio between building height and road width is small."*

[Figure]

**Figure S16.** Diurnal cycle of near surface air temperature simulated with different model set-ups. "Tdefault" indicates that the simulation uses the default calculation of surface temperature in WRF, while "Tmodified" indicates that the simulation uses the calculation of surface temperature from Li and Bou-Zeid (2014) (which is also used in (Vahmani el al. (2016)). Dots for "Urban_Tdefault_shadow" and "Urban_Tdefault_noshadow" ("Urban_Tmodified_shadow" and "Urban_Tmodified_noshadow") are overlapping at every hour of the day because the simulation results with shadow on/off are very similar.

Secondly, we suggest that the comparison of urbanization impacts versus RMSE in Table 1 is not a robust comparison. Instead, to assess whether urbanization impacts are statistically distinguishable from zero, we added a new statistical analysis to the paper, using the paired Student's t-test with n = 7 days. We did the test to check 1) whether spatially averaged changes in regional meteorology and air quality are statistically significant within the simulation period, and 2) whether spatially resolved changes in regional meteorology and air quality are significant within the simulation period (i.e., for each urban grid cell). For 1), we edited the relevant sentences in the paper that refer to spatial average changes. For 2), we updated all figures with maps in the paper by adding black dots to grid cells with insignificant changes. Please see section 2.5 and section 3 in the paper for these changes. We haven't pasted the changes here because they are distributed throughout our results section, and would take up over 3 pages in this document.

Thirdly, we agree with the reviewer that the validation of urban sites, nonurban sites (shrubs in particular) and the difference between urban versus nonurban sites is important. In the main paper we only showed the validation of urban sites (i.e., we added a sentence to point this out in section 3.1.). Thus, we added a section in the supplemental information (section S2) discussing this topic. In general, our model can capture observations at nonurban sites, and the difference between urban versus nonurban sites. Thus, we believe the results we obtain based on this model set-up are reliable. The text in the paper are as follows (see below).

In the main paper:
*"In addition, we only include observations from monitoring sites that are located in urban grid cells in the Present-day scenario. The validation of near surface air temperatures for both urban and nonurban sites are discussed in section S2 in the supplemental information."*

Please see section S2 in the supplemental information for more details on the validation. We again do not paste it here because it is ~3 pages.

b)     It must be made clear that the simulation with current anthropogenic emissions, but not the city, is a hypothetical one – there cannot be emissions without a city. In the last sentence of the manuscript (lines 570-574), authors say that their results "can be informative for decision making on sustainable urban planning to achieve a balance between climate mitigation/adaptation and air quality improvements". Honestly, I do not see how. This type of studies may have a scientific value, in the sense that they demonstrate the importance of taking into account the presence of the city in the simulation of air quality and meteorology (it would be interesting to see if the simulation with the city provides better results compared to measurements than the simulation without the city). But I do not see how they can be helpful for urban planning. Replacing the city with shrubs cannot certainly be considered a strategy to manage urban climate or improve air quality. The differences that authors estimated between the urban and the no-urban simulations are not the maximum difference that can be obtained managing the land use. They actually do not give any information about the impact of any realistic mitigation strategy based on land use management. I think it is very important that authors clarify what they have in mind because this is at the basis of the motivation of the whole manuscript.

We thank the reviewer for bringing this up. To emphasize that the "Nonurban" simulation is a hypothetical scenario, we added the following sentences to section 1, and revised sentences in section 2.5 and section 4.

Section 1
*"In this paper, we aim to quantify the importance of historical land cover change on air pollutant concentrations, and thus the "Nonurban" scenario assumes current anthropogenic pollutant emissions. This hypothetical scenario cannot exist in reality, since current anthropogenic emissions would not exist without the city, but our intent is to tease out the relative importance of land cover change through urbanization (assuming constant emissions) on air pollutant concentrations."*

Section 2.5
*"Note that all three aforementioned scenarios adopt identical anthropogenic emission inventories described in Section 2.3. Using current anthropogenic emissions for "Nonurban" is a hypothetical scenario that cannot exist in reality, but allows us to tease out the effects of land surface changes via urbanization on meteorology and air pollutant concentrations."*

Section 4
*"The two main simulations of focus in this study are the real-world "Present-day" and the hypothetical "Nonurban" scenarios…"*

We agree with the reviewer that the last sentence of the manuscript might not be an appropriate implication of the findings in this study. Therefore, we deleted that sentence, and added the following paragraph to the manuscript in section 4.

*"This study highlights the role that land cover properties can have on regional meteorology and air quality. We find that increases in evapotranspiration, thermal inertia, and surface roughness due to historical urbanization are the main drivers of regional meteorology and air quality changes in Southern California. …Our findings indicate that air pollutant concentrations have been impacted by land cover changes since pre-settlement times (i.e., urbanization), even assuming constant anthropogenic emissions. These air pollutant changes are driven by urbanization-induced changes in meteorology. This suggests that policies that impact land surface properties (e.g., urban heat mitigations strategies) can have impacts on air pollutant concentrations (in addition to meteorological impacts); to the extent possible, all environmental systems should be taken into account when studying the benefits or potential penalties of policies that impact the land surface in cities."*

Detailed comments:
1) Lines 64-66. Urban regions in semi-arid or arid surroundings have a weak (or non- existent) daytime UHI, but they have a very strong nocturnal UHI. I think authors missed the fundamental difference between daytime and nighttime UHI, (being the latter the most frequent).

Thank you for catching this mistake. We modified the sentence as follows.

*"In particular, urban regions built in semi-arid or arid surroundings tend to have a weak daytime UHI or even a UCI, whereas those built in moist regions tend to have a larger daytime UHI (Fan et al., 2017; Peng et al., 2012)."*

2) Line 168. On which basis authors claim that the period chosen is representative of summer conditions in Southern California?

Typical summer days in Southern California are clear or mostly sunny days without precipitation. The chosen simulation period has these characteristics. We added a figure in the supplemental information (Figure S8) showing the diurnal cycle of averaged (observed) near surface air temperature over JJA (June, July and August) and over our simulation period. We also added a sentence in the main paper.

*"This simulation period is chosen as representative of typical summer days in Southern California, which are generally clear or mostly sunny without precipitation. A comparison of observed diurnal cycles for average near surface air temperatures over JJA (June, July and August) versus over our simulation period is shown in Figure S8 in the supplemental information."*

[Figure]

**Figure S8.** Diurnal cycles for observed near surface air temperature (K) over JJA (June, July and August) in blue, and over our simulation period in yellow. Observations are obtained from MesoWest (https://mesowest.utah.edu/), which are available at Mesonet API (https://developers.synopticdata.com/mesonet/). Mean values are derived by averaging over all observational sites available for the innermost domain and the aforementioned period for each hour of day. Orange and grey curves show the maximum and minimum air temperature at each hour of the day for JJA. Results show that our simulation period (July 1-7) is representative of summertime meteorology for our domain.

3) Line 174. Please provide the value of the depth of the lowest model level.

Thanks for the suggestion. We added it to the manuscript.

*"The average depth of the lowest model level is 53 m for all three domains."*

4) Line 215. Is the irrigation module implemented just for the pervious fraction of the urban cells, or also for the rural cells (to account for agricultural crops in the region)?

The irrigation module is just for the pervious fraction of the urban grid cells. The rural grid cells surrounding urban regions in Southern California are mostly natural land cover (e.g., shrub lands), and do not need to take irrigation into account. We added this information to the related sentence.

*"Here we use an irrigation module developed by Vahmani and Hogue (2014), which assumes irrigation occurs three times a week at 2100 PST on the pervious fraction of urban grid cells."*

5)    Line 302. I would avoid indicating the percentage for temperature. This would depend on the unit (if you use Celsius or Kelvin). I would just put degrees.

Thanks for the suggestion. We have gone through the manuscript and changed all expressions of temperature in percentage to absolute values in Kelvin.

6)    Line 303. On which basis authors claim that this is "acceptable".

We agree with the reviewer that "acceptable" is somewhat vague and unclear, so we modified the sentence as below, and added the comparison between our evaluation results and recommended model performance benchmarks to the supplemental information.

*"The statistical results indicate that while model simulations underestimate near-surface air temperature, $O_3$ and $PM_{2.5}$ concentrations by 1.0 K, 22% and 31%, respectively. The comparison between our evaluation results and recommended model performance benchmarks is presented in supplemental information Table S2."*

7)    Section 3.2.3. I suggest studying the difference in sea breeze front progression between the two cases (urban and no-urban). This will give a better understanding of what is happening.

[Figure]

**Figure R1.1** Sea breeze flow in the Los Angeles basin. Note the two sea breeze fronts formed at San

Fernando convergence zone (upper left), and the Elsinore convergence zone (lower right). (Figure adopted from: https://www.aviationweather.ws/097_Sea_Breeze_Soaring.php)

Figure R1.1 shows the two major sea breeze convergence phenomena in Los Angeles basin occurring in the San Fernando convergence zone and the Elsinore convergence zone. They are both formed by the meeting of two sea breezes that had flowed around the mountains. The sea breeze that flows across the Los Angeles coastal plain extends into the Mojave Desert. The sea breeze front of this branch of flow is not shown in our innermost domain. Therefore, we present here the difference in the wind vectors in the lowest model layer from 10 am to 5 pm instead of the progression of sea breeze front (Figure R1.2). The wind vectors are pointing towards the sea, which indicates that there is a significant decrease in wind speed in all hours of day presented here.

[Figure]

**Figure R1.2** Wind vector plots for differences (Present-day – Nonurban) in temporally averaged wind speed in the lowest model layer from 10 am to 5 pm. The background plot shows the differences (Present-day – nonurban) in air temperature.

8)    Lines 335-338. This is not clear. Before it is said that urbanization decreases temperatures and not increases.

The averaged temperature in the urban region decreases during the morning and afternoon. However, temperature changes vary spatially. While reductions in air temperature occur across the urban region during the morning, and in the inland urban region during the afternoon, temperatures actually increase in the coastal region during the afternoon. We specified the time when increases in temperature occur in the sentence.

*"Note that the onshore sea breeze decreases in strength despite higher temperatures in the coastal region of Los Angeles during the afternoon, which would tend to increase the land-sea temperature contrast and thus be expected to increase the sea breeze strength."*

9)    Line 370. During nighttime atmosphere cools. The energy stored in the building during daytime (what authors call upward ground heat flux, I suppose) reduces the cooling. The higher PBL in the

urban simulation will reduce the cooling too because the effect of the surface cooling is distributed in a greater depth than in the no-urban case. The two mechanisms (energy stored in buildings, and high PBL), both reduce cooling. They do not compete they go in the same direction.

Thank you for catching this mistake. We revised the related sentences in section 3.2.4.

*"The climate response to urbanization during nighttime is driven by the combined effects of (a) temperature increases from increasing upward ground heat fluxes, and (b) temperature increases from increasing PBL heights. Increased soil moisture (from irrigation) and use of man-made materials leads to higher thermal inertia of the ground; this in turn leads to increased heat storage during the day and higher upward ground heat fluxes and thus surface temperatures at night. Increasing PBL heights can also lead to warming because of lower air cooling rates during nighttime. Changes in PBL heights are associated with surface roughness changes since shear production dominants TKE at night. Coastal (inland) regions show larger (smaller) variation in roughness length (Figure 2e), which leads to larger (smaller) increases in PBL heights (Figure S14c). Despite larger increases in PBL heights in coastal versus inland regions, smaller air temperature increases occur in coastal versus inland regions, likely due to accumulative effects from coastal to inland regions with onshore wind flows."*

10) Lines 375-376. Same as above, during the night there is not heating, there is cooling.

Please refer to our response to comment 9.

---

## Author Comment (AC2) · 13 Feb 2019

**Response to Anonymous Referee #2**
(Note: Reviewer comments are listed in grey, and responses to reviewer comments are in black. Pasted text from the new version of the paper is in italics.)

Dear authors, the paper is well written and clearly structured, however I would recommend a number of major changes in order to be suitable for publication. Please find my comments below:

We thank the reviewer for his/her thoughtful and valuable comments. These comments substantially help to improve our manuscript by addressing these issues.

General: I see a general problem in the definition of the scope of the study. A 'before human settlement' scenario should not consider emissions at all and further describes a period about 100-150 years ago which means that you would also have to consider a different climate period, land use etc.. I definitely would recommend to redefine the scope of the study, because in the current state, just distinguishing between 100% urban vs. 0% urban is not sufficient to analyze the above mentioned scenario.

We thank the reviewer for bringing this up. The "Nonurban" simulation in this study is a hypothetical scenario in which we assume current anthropogenic emissions and climate, but natural land cover prior to human perturbation. By doing so, we focus our study on the relative importance of land cover changes via urbanization on regional meteorology and air quality. To make it clearer in the paper that the "Nonurban" simulation is a hypothetical rather than realistic scenario, we added the following sentences to the introduction (section 1), and modified relevant sentences in section 2.5 and the conclusion (section 4).

Section 1
*"In this paper, we aim to quantify the importance of historical land cover change on air pollutant concentrations, and thus the "Nonurban" scenario assumes current anthropogenic pollutant emissions. This hypothetical scenario cannot exist in reality, since current anthropogenic emissions would not exist without the city, but our intent is to tease out the relative importance of land cover change through urbanization (assuming constant emissions) on air pollutant concentrations."*

Section 2.5
*"Note that all three aforementioned scenarios adopt identical anthropogenic emission inventories described in Section 2.3. Using current anthropogenic emissions for "Nonurban" is a hypothetical scenario that cannot exist in reality, but allows us to tease out the effects of land surface changes via urbanization on meteorology and air pollutant concentrations."*

Section 4
*"The two main simulations of focus in this study are the real-world "Present-day" and the hypothetical "Nonurban" scenarios"*

I am further not fully convinced about the added benefit of this study for sustainable urban planning

recommendations. I am aware that these model systems are not suitable for applied urban planning, but however the currently existing urban canopy models in WRF-Chem (and other models), together with high resolution datasets for both emission and urban morphology do offer a framework for a number of different scenarios in the context of climate change/UHI mitigation. Recent studies have been analyzing the impact of highly reflecting building materials, urban greening or varying building density for a number urban areas. These aspects should also be possible with this model system and worth being discussed in order to increase the scientific substance of that work and highlight the new contribution to the field. In light of the scope of the journal, it should also be worked out more detailed what are the implications for atmospheric science in general rather than purely investigating local/regional aspects.

We agree with the reviewer that our description of the implications of this study were somewhat ambiguous in the submitted version of the paper. The main point that we intend to make here is that land surface changes on their own can have a significant influence on regional air quality via altered meteorological conditions. Therefore, we should consider the benefits and penalties of UHI mitigation strategies (i.e., since most of them modify land surface properties) from the viewpoint of both climate and air quality to achieve a comprehensive assessment. We revised the conclusion section (section 4) as follows:

*"This study highlights the role that land cover properties can have on regional meteorology and air quality. We find that increases in evapotranspiration, thermal inertia, and surface roughness due to historical urbanization are the main drivers of regional meteorology and air quality changes in Southern California. ...Our findings indicate that air pollutant concentrations have been impacted by land cover changes since pre-settlement times (i.e., urbanization), even assuming constant anthropogenic emissions. These air pollutant changes are driven by urbanization-induced changes in meteorology. This suggests that policies that impact land surface properties (e.g., urban heat mitigations strategies) can have impacts on air pollutant concentrations (in addition to meteorological impacts); to the extent possible, all environmental systems should be taken into account when studying the benefits or potential penalties of policies that impact the land surface in cities."*

I am convinced, that the model system, combined with the emission and land surface data sets offer a promising tool for discussing air quality/meteorology interactions in large urban areas such as Los Angeles, but however think that the variety of scenarios should be increased in order to allow for a more robust results towards currently relevant issues. The authors rely mostly on previous work with equal model configuration. Therefore, the own contribution to the field and the new development does not come out clearly. The paper however is well written and easy to follow, but crucial points have to be considered in a review before being able undertake a detailed line-by-line evaluation.

The study certainly builds on our prior work, but this paper focuses on air quality impacts, whereas our previous research was on only meteorology. Thus, the most important contribution of this work is that we investigate a totally different environmental system than previous work. In order to do so, we also add a new modeling component (atmospheric chemistry) that is not presented in past work.

Other smaller additions compared to our past work is that we turn on the shadow model and incorporate GIS-based building morphologies, which make the model simulations more representative of current day weather conditions in LA. Moreover, while the influence of land use changes on regional weather has been well studied, its influence on regional air quality has been seldom studied with accurately resolved land surface data, especially in the Southern California region. Therefore, our study fills this research gap. We added several sentences in the last paragraph of introduction section to emphasize this point.

*"Note that this paper builds on our prior study Vahmani et al. (2016), but focuses on air quality impacts, whereas our previous research was on meteorological impacts only. While the influence of land surface changes on regional weather has been investigated in numerous past studies, its influence on regional air quality has been seldom studied in past work."*

Moreover, the focus of this study is on the impact of land surface changes on regional meteorology and air quality. Thus, the two major scenarios discussed are "Nonurban" and "Present-day" scenarios, which characterized land surface prior to human perturbation and under current conditions respectively. We also included a supplemental scenario "Present-day No Irrigation" that teases out the effects of irrigation.

1. The scope of the study should be defined more clearly in light of the above mentioned points. The experimental design should be expanded, in order to include more own ideas/developments.

As mentioned in our previous responses, we focus this study on the relative importance of land cover changes via urbanization on regional meteorology and air quality, and assume identical climate and anthropogenic emissions in both scenarios. In our simulations, we implemented real-world representation of land surface properties in the "Present-day" scenario, which made it possible to tease out the most important land surface factors. Our results indicate that land surface changes have a significant influence on regional air quality via altered meteorological conditions. This suggests that policies that impact land surface properties should take all environmental systems into account when studying the benefits or potential penalties of the policies. We feel that this is a solid focused story for the paper, and adding additional simulations would only dilute the main points we are trying to make. In other words, adding more complexity to the study would only muddle the story.

2. One interesting and highly relevant point in my opinion is the 'irrigation' module which might offer a nice tool for testing different irrigation scenarios.

We agree that the proposed research idea is an interesting topic. However, it would be more appropriate as an individual study on the influence of irrigation on regional climate and air quality. This isn't the main research question we are trying to answer. In this study, we want to keep the scope well defined in answering the posed research questions on how historical land surface changes have affected regional climate and air pollutant concentrations in Southern California. Thus, investigating the regional influence of irrigation sounds interesting but beyond our motivation and scope.

3. Why did you select a single-layer urban canopy model rather that a more complex multi-layer canopy representation (BEP/BEM)? The latter should deliver higher accuracy close to the ground I guess? What is the depth of the lowest model level?

As suggested by Kusaka et al. (2001) , the model performance of UCM and BEP/BEM with regard to studying mesoscale heat islands are comparable. Chen et al. (2011) also mentions that the UCM may be more suitable than BEP/BEM for weather and air quality prediction. In addition, coupling BEP/BEM to WRF/Chem would be an extremely complex model development exercise,    and the resulting model would be prohibitively computationally expensive, but for likely little additional benefit in the quality of simulations. Therefore, we choose to couple the UCM instead of BEP/BEM to WRF/CHEM.

The averaged depth of the lowest model level is 53 m for all three domains. This information has been added to the paper at the last sentence in section 2.1.

*"The average depth of the lowest model level is 53 m for all three domains."*

4. Where do the input parameters for SLUCM come from?

We use NLCD impervious surface data for impervious fraction of each grid cell. For surface albedo of roof, building wall, and road, we assign the grid cell albedo value derived from MODIS. Building morphologies (including building height, standard deviation of roof height, building width and road width) are from NUDAPT where available. Where NUDAPT data are unavailable, we adopt average building and road morphology from LARIAC. This information is mentioned in section 2.2. For the other parameters in UCM (e.g., anthropogenic latent heat, surface emissivity), we use default WRF settings documented in file URBPARM.TBL. We added this information to section 2.2.

*"For the other parameters in the UCM (e.g., anthropogenic latent heat, surface emissivity), we use default WRF settings documented in file URBPARM.TBL."*

5. What is the additional gain of a 30 m land surface classification which has to be scaled to 2 km model resolution?

We chose to use 30 m-resolution 33-category NLCD mainly for two reasons. First, urban land use varies at spatial scales on the order of 10s of meters. So it works best to define land use at spatial scales of 10s of m, and then aggregate to the model grid resolution. It would be difficult to detect land use using data at 2km resolution. Second, the 30 m-resolution land use dataset has 33 categories of land use type, which divides urban type into three sub-types: low-intensity residential, high-intensity residential, and industrial/commercial. This allows different parameterizations for different sub- urban types, which better characterize land surface properties.

6. Is there a problem with regard to the discrepancy between emission inventory and model resolution?

No, there should not be a problem. The resulting air quality predictions are simply lower resolution than they would be if they were at 2km. We ensured that the total emissions within in the domain are kept consistent after regriding.

7. How realistic is the surrounding 'non-urban' land use classification for the 'historical' scenario?

The dominant natural land cover type surrounding Los Angeles and San Diego metropolitan areas is shrub. So it is reasonable to assume shrub as the land use type in the "Nonurban" scenario. The land surface properties of these grid cells in the "Nonurban" scenario are derived using the inverse distance weighting approach, which is mentioned in section 2.5, and consistent with our previous publication.

*"For the Nonurban scenario, we assume natural land cover prior to human perturbation, and replace all urban grid cells with "shrubs" (Figure 1c). We modify MODIS-retrieved albedo, GVF and LAI in these areas based on properties for shrub lands surrounding urban regions in the Present-day scenario. A detailed explanation on this method (inverse distance weighting approach) can be found in Vahmani et al. (2016)."*

8. How well does the model simulate urban AND rural parameters?

In section 3.1, we showed how well the model simulated urban variables (i.e., near surface air temperature, and pollutant concentrations). We agree with the reviewer that it is also important for the model to capture nonurban (especially shrub) air temperatures well, thus we included a discussion on this topic in section S2 in the supplemental information. Note that nonurban observational sites that measure pollutant concentrations are rare. Thus, we decided not the discuss how well the model simulated pollutant concentrations in nonurban area. The new text in the main paper is pasted below. Please see section S2 in the supplemental information for more details on the validation. We did not paste it here because it is ~3 pages.

In the main paper:
*"In addition, we only include observations from monitoring sites that are located in urban grid cells in the Present-day scenario. The validation of near surface air temperatures for both urban and nonurban sites are discussed in section S2 in the supplemental information."*

9. Please specify how results from this study can serve as contribution for applied urban planning.

As mentioned in our response to your second general concern, we changed the last paragraph in the conclusion section, which explains the implications of our study. Please see that response for more detail.

10. In relation to other chapters, the introduction is slightly too long. Try to focus on the relevant points here and shorten were possible.

We think that the background knowledge, brief literature review, and research gaps described in the introduction section are necessary for a clear explanation of the scope and motivation of this study.

The flow of the introduction section is as follows. First, we point out that urbanization has led to profound modification of the land surface. We then explain how changes in land surface properties can affect regional meteorological fields such as surface and air temperature, wind speed and PBL height. We go on to demonstrate how those changes in meteorology due to land surface modification can in turn affect air pollutant concentrations via different mechanisms. While there are a number of studies that have investigated the impacts of land surface changes on regional meteorology, limited studies have quantified the impact of land surface changes on regional air quality, especially for the Southern California region, which has a history of severe air pollutant problems. In addition, recent studies have made it possible to utilize satellite land surface data in model simulations, which better predict regional weather in urbanized regions, and urban versus nonurban differences. Thus, our study adopts the modified model configuration, and aims to characterize the influence of historical urbanization on urban meteorology and air quality in Southern California.

Please find below comments for specific sections, which partly have been addressed in the main points above.
Ln 11: ventilation not a good expression here

We think that "ventilation" is a proper expression here because it appropriately describes the ability of atmosphere to transport pollutants out of the studied area.

13: 'before human settlement' is a bit misleading here, as it is not entirely captured by your model setup. As mentioned before, some effort has to be put in a clear definition of the scope of your study. What problem should be addressed – also in light of recommendations for real urban planning (Lines 570-573?

The two concerns mentioned in this comment are addressed in the first two general comments respectively.

43: "Differences in surface temperature..." What was the purpose of these studies mentioned here and what do they try to answer? How does this sentence relate to your study and the intention for this work?

The UHI and UCI represents urban versus nonurban difference in surface or air temperature. They are both climate phenomena at urban scale that occur due to variability in land cover changes. Here we summarize possible ways in which land surface modifications can affect surface/air temperature difference between urban and nonurban areas, which is what causes the UHI/UCI. The temperature difference between the "Present-day" scenario and "Nonurban" scenario discussed in our study is analogous to the UHI/UCI. Thus, the background information here is necessary.

47: "UCI": How does this relate to your study?

This is explained in the response to the comment above.

67: What is the role of the atmospheric aerosol burden for UHI formation?

The role of atmospheric aerosol on UHI intensity is an active research topic, and yet no consensus has been reached. For example, Kumar et al. (2017) carried out a Global Climate Model simulation, and suggested that daytime cooling (UCI) can be partially attributed to absorbing aerosols over Indian cities. Cao et al. (2016) used satellite observations, and found positive correlation between urban–rural difference in AOD and nighttime UHI.

73: better "characteristics/shape of the PBL is dependent on..."

Changed. Thanks!

Changed. Thanks!

86: unclear what is meant by "meteorological changes via altered emissions,..."

We changed the sentence as follows:

*"Meteorology can affect emission rates, chemical reaction rates, gas-particle phase partitioning of semi-volatile species, pollutant dispersion, and deposition; thus, it plays an important role in determining air pollutant concentrations."*

115: Why do higher PBLs increase $PM_{2.5}$ concentration? Please discuss the related processes here.

In our text, we mentioned that Chen et al. (2018) found that higher PBLs decrease $PM_{2.5}$ concentrations. Please find the original sentence below:

*"Chen et al. (2018) studied urbanization in Beijing, and found that modification of rural to urban land surfaces has led to increases in near-surface air temperature and PBL height, which in turn led to increases (+9.5 ppb) in surface $O_3$ concentrations and decreases (–16.6 $\mu g/m^3$) in $PM_{2.5}$ concentrations."*

119: How exactly does your experimental setup treat the "wide heterogeneity of urban land surface processes" compared to existing studies? A large number of studies already exist using model systems (e.g. WRF) which include urban canopy models with varying complexity (SLUCM, BEP), which consider a similar level of heterogeneity than your experiments? Please discuss your statement.

While previous studies have used models with different levels of complexity, most of them failed to incorporate real-world land surface property data as input. They used default WRF settings for land

surface properties such as building morphology, albedo, vegetation fraction, which either is out of date, or lacks spatial heterogeneity. By contrast, in this study we use NLCD for land cover type and impervious fraction, satellite-retrieved data for albedo, vegetation fraction and leaf area index, and GIS-based data for building morphology, which resolves spatial heterogeneity of land surface properties, and better predicts regional weather and air quality. The default version of the WRF/UCM assumes that many land cover properties are spatially homogeneous, which is not realistic.

122: unclear expression "amongst"?

We changed the sentence as follows:

*"In addition, only few studies investigate interactions between land surface changes and air quality for the Southern California region (Taha, 2015; Epstein et al., 2017; Zhang et al., 2018b), which is one of the most polluted areas in the United States (American Lung Association, 2012)."*

134-140: It should be made clear which new aspects you aim to analyze compared to the studies mentioned above. In my opinion simply turning urban on/off does not reveal significantly new insights. Further the term "human disturbance" is unclear, as this would also involve air quality modifications.

As we mentioned in the introduction, there are limited studies on the effect of land surface change via urbanization on regional air quality, most of which do not resolve the real-world spatial heterogeneity. In addition, there are several recent studies by our group, which incorporate satellite data for land surface characterization within Southern California, and quantifies the effect of land surface changes on regional climate including temperature and wind speed. Thus, this study combines the research idea of these two types of studies together, and aims to characterize the influence of land surface changes via historical urbanization on urban meteorology and air quality in Southern California using highly resolved land surface characterization.

We focus on the land surface modifications from human disturbance in this study, and use specific phrasing about this in the paper.

Abstract
*"In this study we characterize the influence of land surface changes via historical urbanization from before human settlement to present-day on meteorology and air quality in Southern California using the Weather Research and Forecasting Model coupled to chemistry and the single-layer urban canopy model (WRF/Chem-UCM)."*

Last paragraph in introduction section
*"Therefore, this study aims to characterize the influence of land surface changes via historical urbanization on urban meteorology and air quality in Southern California by comparing a "Present-day" scenario with current urban land surface properties and land surface processes to a "Nonurban" scenario assuming land surface distributions prior to human perturbation."*

Section 2.5
*"For the Nonurban scenario, we assume natural land cover prior to human perturbation, and replace all urban grid cells with "shrubs" (Figure 1c)."*

First paragraph in conclusion section
*"In this study, we have characterized the impact of land surface changes via urbanization on regional meteorology and air quality in Southern California using an enhanced version of WRF/Chem-UCM. ... The two main simulations of focus in this study are the real-world "Present-day" and the hypothetical "Nonurban" scenarios; the former assumes current land cover distributions and irrigation of vegetative areas, while the latter assumes land cover distributions prior to widespread urbanization and no irrigation."*

174: Please specify your lowest model level.

Thanks for the suggestion. We added this information to the manuscript.

*"The average depth of the lowest model level is 53 m for all three domains."*

175: "process parametrization" unclear

We changed the title to:

*"Land Surface Property Characterization and Irrigation Parameterization"*

180: Please discuss the term "real world representation", answering the question why the WRF default land use classification in WRF is not "real" enough for your case comparing these datasets with your input. What was the idea behind using a 30m dataset? Please briefly discuss the gain of using 30 m land cover data for a maximum resolution of 2 km. How much information "is lost" by the process of "upscaling" the LU data. Would the 2011 NLCD dataset add additional benefit?

By "real world representation", we mean instead of using default land surface properties provided with WRF, we used satellite-retrieved data specifically for the Southern California region. This is beneficial for a better prediction of regional weather.

As we mentioned in response to comment 5, we chose to use 30 m-resolution 33-category NLCD mainly for two reasons. First, urban land use varies at spatial scales on the order of 10s of meters. Second, it separates urban to three sub- urban types, which allows more detailed parameterization.

Also we chose to use the 2006 NLCD dataset in order to keep consistency with previous work from our group.

205: Did you use the additional sub-tiling option in WRF?

No, we didn't. The land surface module (the unified Noah land surface model) we use doesn't have a sub-tiling option. However, the module treats impervious fraction and pervious fraction of the urban grid cell separately.

243: Do you consider daily emission profiles? Meaning, do you find two "peaks" for instance in NOx emission/concentration?

Yes we do. Figure R2.1 shows the diurnal cycle of NOx emissions. The diurnal cycle of NOx concentrations is shown in the paper in Figure 6a. We can see that the emissions of NOx shows two peaks during daytime, and stays high between the two peaks. However, for NOx concentrations, it peaks during morning, and decreases continuously until late afternoon, despite rather high emissions. This indicates that high photolysis rates and high PBL heights due to warm temperatures in the afternoon play an important role on determining NOx concentration during daytime, apart from just emissions.

[Figure]

**Figure R2.1** Diurnal cycle of NOx emissions. Averaged over urban grid cells within simulation period.

267: How realistic is the conversion to shrub-land for all grid cells? Would you expect different effects for a non-urban, but more heterogeneous "before human" land cover?

Please see the response to comment 7.

294: Please indicate better proof of the "good fit" mentioned here. It is not indicated by Figures S1 and 3 for $PM_{2.5}$. How does the correlation coefficient look like? What are the reasons for the poor correlation especially for the high range of the observed concentrations? How representative are the measurement stations? As the ozone concentration is highly dependent on temperature you find a good fit. Does the poor fit for $PM_{2.5}$ relate to high mixing, chemistry, both? How do correlations look like for NO2, NO, CO? Are the simulated diurnal variations realistic? Please also discuss the values from Table 1? Are they particularly good/bad?

We modified the last sentence in section 3.1 as below, and added the comparison between our

evaluation results and recommended model performance benchmarks. The comparison indicates that the evaluation result are close to the ME benchmark for hourly near surface air temperature, and NMB benchmark for hourly Ozone concentrations. For daily $PM_{2.5}$ concentration, the discrepancy between the evaluation and the recommended benchmark is largely due to the underestimation of high observational values. This poor fit at high concentrations is likely occurring due to one or more of the following factors: 1) not including dust emissions in the simulation, which makes up an appreciable fraction of real-world total $PM_{2.5}$, 2) a failure of the emissions inventory to capture high emission rates on particular days, and 3) the chemistry parameterizations in WRF/Chem tending to underestimate $PM_{2.5}$ concentrations at high values, and 4) errors in simulated air pollution meteorology. We also added this information to the main paper.

*"The underestimation of $PM_{2.5}$ concentrations may be occurring due to one or more of the following factors: 1) not including dust emissions in the simulation, which makes up an appreciable fraction of real-world total $PM_{2.5}$, 2) a failure of the emissions inventory to capture high emission rates on particular days, 3) the chemistry parameterizations in WRF/Chem tending to underestimate $PM_{2.5}$ concentrations at high values, and 4) errors in simulated air pollution meteorology. Table 1 shows four statistical metrics for model evaluation, including mean bias (MB) and normalized mean bias (NMB) for the quantification of bias, and mean error (ME) and root mean square error (RMSE) for the quantification of error. The statistical results indicate that model simulations underestimate near-surface air temperature, $O_3$ and $PM_{2.5}$ concentrations by 1.0 K, 22% and 31%, respectively. The comparison between our evaluation results and recommended model performance benchmarks is presented in the supplemental information Table S2."*

The correlation coefficients for near surface air temperature, $O_3$ concentration and $PM_{2.5}$ concentration are 0.92, 0.82, 0.025 respectively. The observation sites for air temperature, $O_3$ concentration, and $PM_{2.5}$ concentration are shown is Figure S9. The sites are spread across the urban region in the model domain, and should be representative of the urban region in Southern California. On the other hand, point measurements do not capture the same spatial footprint as 2 km model grid cells. Thus, some model versus observational discrepancy is always expected, making interpretation difficult.

Figure S11 and Figure S12 shows the comparison between observed and modeled diurnal cycle for near surface air temperature and $O_3$ concentrations. Values for each hour are averaged over the whole simulation period for all observation sites. The results indicate that while model underestimates both observed air temperature and $O_3$ concentrations, it follows the diurnal pattern well. These figures are in the supplemental information, and we added a sentence in the main paper.

*"…(Comparisons between observed and modeled diurnal cycles for near surface air temperatures and O3 concentrations are also presented in the supplemental information, Figure S11 and S12.)"*

[Figure]

**Figure S11.** Diurnal cycle of observed and modeled near surface air temperature.

[Figure]

**Figure S12.** Diurnal cycle of observed and modeled surface O₃ concentrations (ppb).

347: Can you find impacts on the strength of the sea breeze when there is no urban area left?

The Present-day versus Nonurban difference in the strength of sea breeze is shown by Figure S14 in the supplemental information. Land surface changes via urbanization has led to decrease in wind speed throughout the day due to increase in land surface roughness. This weakening is more significant during the day, especially in the afternoon, when the baseline sea breeze is strongest.

[Figure]

**Figure S14.** Spatial patterns of differences (Present-day – nonurban) in temporally averaged values during morning, afternoon and nighttime for (a,b,c) PBL heights, and (d,e,f) averaged wind speed under within PBL. Note that values are shown only for urban grid cells. Morning is defined as 7 PST to 12 PST, afternoon as 12 PST to 19 PST, and nighttime as 19 PST to 7 PST. Note that values are shown only for urban grid cells.

363: What is the order of difference between shadow model on/off?

There are two major differences in model configuration between this study and our previous publication (Vahmani and Ban-Weiss, 2016; Vahmani et al., 2016). First, in this study, we turn on the shadow model, while our previous study doesn't account for the shadow effect. Second, we use the model default calculation of surface temperature (which will affect the calculation of air temperature), while our previous study uses an alternative calculation of surface temperature. After a careful comparison among different model set-ups, we found that the parameterization of surface temperature is the more important factor that affects daytime air temperature (Figure S16 in the supplemental information). Therefore, we delete "and shading effects within urban canopies" in the sentence "In addition, increases in thermal inertia caused by use of manmade materials (e.g., pavements and buildings) and shading effects within urban canopies can contribute to simulated temperature reductions during the morning.". We also modified the last paragraph in section 3.2.3 as below.

*"Note that changes in air temperature during daytime shown here disagree with Vahmani et al. (2016). While our study detects daytime temperature reductions due to urbanization, Vahmani et al. (2016) suggests daytime warming. After detailed comparison of the simulations in our study versus Vahmani et al. (2016), we find that the differences are mainly associated with UCM configuration. First, our study uses model default calculations of surface temperature for the impervious portion of urban grid cells, whereas Vahmani et al. (2016) applied an alternative calculation proposed by Li and Bou-Zeid, 2014. Li and Bou-Zeid, 2014 intended the alternate surface temperature calculations to be performed as a post-processing step rather than during runtime. After a careful*

*comparison among different model set-ups, we found that the parameterization of surface temperature is an important factor that affects simulated daytime air temperature (See Figure S16). Second, our study accounts for shadow effects in urban canopies, whereas Vahmani et al. (2016) assumes no shadow effects. (We note here that the default version of the UCM has the shadow model turned off. The boolean SHADOW variable in module_sf_urban.F needs to be manually switched to true to enable the shadow model calculations. With the shadow model turned off, all shortwave radiation within the urban canopy is assumed diffuse.) We suggest that it is important to include the effects of building morphology on shadows within the canopy, and to track direct and diffuse radiation separately, and therefore perform simulations in this study with the shadow model on. Note that the effect of shadows is not as significant as the parameterization of surface temperature for most of the domain in our study because the ratio between building height and road width is small."*

[Figure]

[Figure]

**Figure S16.** Diurnal cycle of near surface air temperature simulated with different model set-ups. "Tdefault" indicates that the simulation uses the default calculation of surface temperature in WRF, while "Tmodified" indicates that the simulation uses the calculation of surface temperature from Li and Bou-Zeid (2014) (which is also used in (Vahmani el al. (2016)). Dots for "Urban_Tdefault_shadow" and "Urban_Tdefault_noshadow" ("Urban_Tmodified_shadow" and "Urban_Tmodified_noshadow") are overlapping at every hour of the day because the simulation results with shadow on/off are very similar.

367: Origin of the UCM parameters?

Please refer to our response to the fourth comment.

381: Calculation of the ventilation coefficient?

We added the calculation of ventilation coefficient to the main content in section 3.2

*"... The integral form of this calculation can be written as (Eq1).*

$$Ventilation\ Coefficient = \int_0^{PBL\ height} U(z)\ dz \qquad (Eq1)$$

*Given that the atmosphere is stratified in models, Eq1 can be discretized as Eq2:*

$$Ventilation\ Coefficient = \sum_{i=1}^m U(z_i) \times \Delta z_i \qquad (Eq2)$$

*Where $U(z_i)$ stands for horizontal wind speed within the $i^{th}$ model layer (m/s), $\Delta z_i$ is the depth of*

*$i^{th}$ model layer that is within PBL (m), and m is the number of vertical layers up to PBL height."*

386: Please evaluate the quantity values here? Provide relative numbers.

Thanks for the suggestion. We add relative values to the sentence.

*"... the spatially averaged decreases are –726 m²/s (–23%) and –560 m²/s (–34%), respectively."*

395: What is the relation between PBL height and surface roughness? Please provide more details. Can you find proof for this in your study?

We've discussed how surface roughness affects PBL height in the third paragraph in the introduction section. The nighttime PBL height is associated with variations in wind speed, which is related to variations in surface roughness. By changing shrubs (homogeneously throughout the urban region) to buildings (heterogeneously varies according to sub-urban types), the variation in surface roughness is increased. We modified the relevant sentence in section 3.2.4.

*"Coastal (inland) regions show larger (smaller) variation in roughness length (Figure 2e), which leads to larger (smaller) increases in PBL heights (Figure S14c)."*

490: Can you say something about the change of PBL dynamic comparing urban and non-urban. I suspect concentration of $PM_{2.5}$ is highly dependent on the boundary layer depth. Expecting lower PBLs in "urban-free" areas actually should decrease $PM_{2.5}$ in summer?

As we discussed in section 3.2.5, air temperature (surface roughness) changes are the major driver of PBL changes during the day (night) for urban grid cells. While land surface properties don't change among nonurban grid cells (i.e., outside the urban domain), changes among urban grid cells will affect nonurban grid cells via transport of moisture, energy and momentum. Thus, most nonurban regions show similar trends for changes in PBL height compared to urban regions (discussed in the response to the next comment).

Responding to your last sentence, lower PBLs would lead to greater $PM_{2.5}$ concentrations, not lower concentrations.

530: What happens to the PBL height in non-urban environment? Even deeper?

Figure R2.2 shows changes in PBL height for Present-day – Nonurban (showing values only for grid cells that are not deemed urban in the Present-day scenario). PBL height decreases in most regions during the day (i.e., morning and afternoon), while changes at night are negligible. The tendency of changes in these grid cells outside the urban region are similar to that in urban grid cells.

[Figure]

**Figure R2.2** Spatial patterns of differences (Present-day – nonurban) in temporally averaged values during morning, afternoon and nighttime for (a,b,c) PBL height. Note that values are shown only for grid cells that are not deemed urban in the Present-day scenario.

535: Specify "enhanced".

We added the following sentence to the paper to specify "enhanced".

*"We use satellite data for the characterization of land surface properties, and include a Southern California-specific irrigation parameterization."*

541: How confident are you that the land use class in the "before-human" settlement is correct? Or is it just a guess?

Please refer to the response to comment 7.

573: As mentioned earlier I am not entirely convinced, how findings from this study could be used for applied urban planning? You mention 'mitigation and adaptation', but a complete 'removal' of the urban area should be hard to transfer into an actual applicable strategies. Maybe more 'moderate' scenarios would be better. However, avoiding a complete re-doing of model experiments, the scope of the study should be formulated differently

Please refer to our responses to the second general comment, and the first detailed comment.

**Reference**

Cao, C., Lee, X., Liu, S., Schultz, N., Xiao, W., Zhang, M. and Zhao, L.: Urban heat islands in China enhanced by haze pollution, Nat. Commun., 7, 12509, 2016.

Chen, F., Kusaka, H., Bornstein, R., Ching, J., Grimmond, C. S. B., Grossman-Clarke, S., Loridan, T., Manning, K.W., Martilli, A., Miao, S. and Sailor, D.: The integrated WRF/urban modelling system: development, evaluation, and applications to urban environmental problems, Int. J. Climatol., 31(2), 273–288, 2011.

Kumar, R., Mishra, V., Buzan, J., Kumar, R., Shindell, D. and Huber, M.: Dominant control of agriculture and irrigation on urban heat island in India, Sci. Rep., 7(1), 14054, 2017.

Kusaka, H., Kondo, H., Kikegawa, Y. and Kimura, F.: A simple single-layer Urban Canopy Model for atmospheric models: comparison with multi-layer and slab models, Bound.-Layer Meteorol., 101(3), 329–358, 2001.

Vahmani, P. and Ban-Weiss, G. A.: Impact of remotely sensed albedo and vegetation fraction on simulation of urban climate in WRF-urban canopy model: A case study of the urban heat island in Los Angeles. J. Geophys. Res. Atmos., 121(4), 1511–1531, 2016.

Vahmani, P., Sun, F., Hall, A. and Ban-Weiss, G. Investigating the climate impacts of urbanization and the potential for cool roofs to counter future climate change in Southern California. Environ. Res. Lett., 11(12), 124027, 2016.

---

## Author Comment (AC3) · 13 Feb 2019

**Response to Anonymous Referee #3**

(Note: Reviewer comments are listed in grey, and responses to reviewer comments are in black. Pasted text from the new version of the paper is in italics.)

This manuscript investigates impacts of urbanization in Southern California on regional meteorology and air quality. Simulations using an innermost domain with 2 km resolution are conducted by WRF-Chem coupled with UCM. The simulations are driven by current climate and anthropogenic emissions with and without urban pixels and are applied to characterize impacts of historical urbanization on regional and temporal distributions of temperature and concentrations of NOx, $O_3$, and $PM_{2.5}$. The authors conclude that urbanization causes daytime decreases in temperature and increases in $O_3$ and $PM_{2.5}$. In the nighttime, the simulation results present nighttime increases in temperature and $O_3$, while the concentrations of NOx and $PM_{2.5}$ show reductions. The authors attribute these changes to urban-induced modifications in various competing drivers including irrigation, thermal properties of building materials and surface roughness.

General comments:

The topic addressed is interesting and relevant to ACP readers. However, I have reservations about the robustness of the conclusions presented. In my opinion, significant revisions with new analysis and more careful model verification of the simulations are required.

We thank the review for his/her helpful comments. We believe that addressing these comments have vastly improved the quality of our paper.

The impact of urbanization is derived from the differences between temperature and concentrations of fields simulated by a WRF-Chem configuration that includes urban pixels and by a scenario where urban pixels were converted to shrub. This methodology has been presented in previous work and the nighttime impact of urbanization has been well documented in the literature. For instance, the paper by Li et al ("Achieving accurate simulations of urban impacts on ozone at high resolution", ERL, 9, 2014) introduced similar configurations (WRF-Chem including anthropogenic emissions, with and without urbanization) and used them to derive impacts of urbanization on air quality by analyzing the differences in the simulated fields between the two scenarios. Although the region and the period of time considered in this manuscript are different, the main idea and the nighttime impact are similar. The daytime impact reported in this manuscript is questionable because its magnitude shows values smaller than the model error (see specific comment 2). Careful analysis of the robustness of the impact is needed, especially given that this impact conflicts with previous results as reported (Line 355). The authors need to emphasize what is new related to this research and how it advances the existing research on the topic.

We thank the reviewer for these comments. The first major idea presented here (robustness of the daytime impact) is also brought up in specific comments 2 and 5. Please see our responses to those comments.

The second major idea is on the novelty of this study. While this study shows some similarity in research idea with previous literature, it extend this research topic in that 1) it includes discussion

on the impact of land surface changes on total and speciated PM$_{2.5}$ concentration, which has been seldom studied, 2) it focuses on the Southern California region where such research is limited but necessary given the high pollutant loads, and 3) it incorporates accurately resolved land surface data. We added a few sentences in the last paragraph of introduction section to clarify these points.

*"... Note that this paper builds on our prior study Vahmani et al. (2016), but focuses on air quality impacts, whereas our previous research was on meteorological impacts only. While the influence of land surface changes on regional weather has been investigated in numerous past studies, its influence on regional air quality has been seldom studied in past work."*

Specific comments:

1. It is unclear why the authors chose a 10-day period of the summer of 2012? And in what basis the period chosen is "representative of typical summer days in Southern California"? Why not using more years?

We chose this 10-day period because the observed meteorology field is representative of typical summer days in Southern California, which are clear (no clouds) and without precipitation. We added a figure in the supplemental information (Figure S8) showing the diurnal cycle of averaged (observed) near surface air temperature over JJA (June, July and August) and over our simulation period. We also added a sentence in the main paper pointing to that figure.

*"This simulation period is chosen as representative of typical summer days in Southern California, which are generally clear or mostly sunny without precipitation. A comparison of observed diurnal cycles for average near surface air temperatures over JJA (June, July and August) versus over our simulation period is shown in Figure S8 in the supplemental information."*

[Figure]

**Figure S8.** Diurnal cycles for observed near surface air temperature (K) over JJA (June, July and August) in blue, and over our simulation period in yellow. Observations are obtained from MesoWest (https://mesowest.utah.edu/), which are available at Mesonet API (https://developers.synopticdata.com/mesonet/). Mean values are derived by averaging over all observational sites available for the innermost domain and the aforementioned period for each hour of day. Orange and grey curves show the maximum and minimum air temperature at each hour of the day for JJA. Results show that our simulation period (July 1-7) is representative of summertime meteorology for our domain.

We do the simulations for year 2012 because it is the most recent year for which an accurate

emissions inventory is available for Southern California.

2. The statistics presented in Table 1 indicate that the impact of urbanization is smaller than the model error for all the fields analyzed. For example, the magnitude of the simulated change in $O_3$ is less than 5.6 ppb (Line 429). The mean and root mean square errors reported in Table 1 are 11.8 ppb and 14.6 ppb, respectively. Thus, the impact described, which is the main conclusion of the manuscript, is not robust given that it lies within the model error. Perhaps, simulations using other years could increase the statistical significance of the results presented.

We suggest that the comparison of urbanization impacts versus model error in Table 1 is not the right comparison. Instead, to assess whether urbanization impacts are statistically distinguishable from zero, we added a new statistical analysis to the paper, using the paired Student's t-test with n = 7 days. We did the test to check 1) whether spatially averaged changes in regional meteorology and air quality are significant within the simulation period, and 2) whether changes in spatial resolved regional meteorology and air quality are significant within the simulation period (i.e., for each urban grid cell). For 1), we edited the relevant sentences in the paper. For 2), we updated all figures with maps in the paper to mark out the insignificant grid cells with black dots, and edited the relevant description of the spatial patterns. Please see section 2.5 and section 3 for those changes. We haven't pasted the changes here because they are distributed throughout our results section, and would take up over 3 pages.

3. The authors state in the conclusion that "…due to historical urbanization are the main drivers of regional meteorology and air quality changes in Southern California" (Line 567). However, the simulations presented in the manuscript cannot be applied to reach such conclusion. There are several critical factors that are not accounted for. For example, the initial and boundary conditions use current atmospheric conditions and therefore do not include the effect of climate change. The amount of the background $CO_2$ concentration specified in WRF is fixed (assuming that both configurations use the same setup except for urbanization as stated). The anthropogenic emissions did not exist before human settlement. I suggest that the authors rephrase their motivation and conclusion, and simply focus on the impact of urbanization without attributing historical changes solely to urbanization.

We agree with the reviewer that the motivation and conclusions were not sufficiently clear in the original paper. Thus, we modified the last paragraph of the introduction section and conclusion section.

*"… In this paper, we aim to quantify the importance of historical land cover change on air pollutant concentrations, and thus the "Nonurban" scenario assumes current anthropogenic pollutant emissions. This hypothetical scenario cannot exist in reality, since current anthropogenic emissions would not exist without the city, but our intent is to tease out the relative importance of land cover change through urbanization (assuming constant emissions) on air pollutant concentrations."*

*"This study highlights the role that land cover properties can have on regional meteorology and air quality. We find that increases in evapotranspiration, thermal inertia, and surface roughness due to*

*historical urbanization are the main drivers of regional meteorology and air quality changes in Southern California. ...Our findings indicate that air pollutant concentrations have been impacted by land cover changes since pre-settlement times (i.e., urbanization), even assuming constant anthropogenic emissions. These air pollutant changes are driven by urbanization-induced changes in meteorology. This suggests that policies that impact land surface properties (e.g., urban heat mitigations strategies) can have impacts on air pollutant concentrations (in addition to meteorological impacts); to the extent possible, all environmental systems should be taken into account when studying the benefits or potential penalties of policies that impact the land surface in cities."*

4. There are some claims that need clarification. For example the authors state in line 152 "In this study, we couple WRF/Chem to the urban canopy model (UCM). . ." However, the WRF/Chem model is already coupled to UCM. I believe what the authors did is activating the option for this coupling. In line 180 "we update the default WRF/Chem to include a real-world representation of land surface physical properties and processes. . ." But again, the options for using NLCD and NUDAPT for land surface representations are available within WRF. Please clarify what is meant by "we update the default WRF/Chem".

Thanks to the reviewer for pointing this out. We modified the sentence "In this study, we couple WRF/Chem to the urban canopy model (UCM). . ." as below.

*"In this study, we activate the urban canopy model (UCM) in WRF/Chem that ..."*

By "update the default WRF/Chem" we mean that we've used GIS-based building morphologies, satellite-retrieved land surface data, and a Southern California specific irrigation module for the simulations, which make the model simulation more representative of current day weather conditions and air quality in Southern California. We also modified gaseous dry deposition in chemistry module based on previous literature so that WRF/Chem can be compatible with 33-category land use types.

5. The ability of WRF-Chem to realistically represent urban processes requires more evaluation to better establish the credibility of the present-day scenario. The comparison between observations and simulations shown in Fig. 3 does not indicate to me a "good fit at lower values" as stated in line 294. The observed low values of temperature are around 290 K, but the simulated temperature shows low values of 287K. The difference between these values is larger that the impact reported. Therefore, better model verification should be considered. I also suggest adding to Fig. 3 panels comparing diurnal variations of observed and simulated temperature, $O_3$ and $PM_{2.5}$ (similar to Fig. 4a).

For the significance of the reported urbanization impact, please refer to our response to comment 2.

Figures S11 and S12 show the comparison between observed and modeled diurnal variations for near surface air temperature and $O_3$ concentrations. (For $PM_{2.5}$ concentrations we use only daily values instead of hourly $PM_{2.5}$ concentrations for reasons that are explained in the supplemental

information section S1; thus, the diurnal variation of observed and simulated PM$_{2.5}$ is not discussed here.) In Figures S11 and S12, values for each hour are averaged over the whole simulation period for all observation sites. The results indicate that while the model underestimates both observed air temperature and O$_3$ concentrations, the shape of the diurnal cycle is well modeled. For air temperature, simulation results tend to capture daytime (relatively higher) values better than nighttime (relatively lower) values. For O$_3$ concentrations, the model predicts lower concentrations better than higher concentrations. Thus, we edited the sentence the reviewer mentioned. And we put these two figures to the supplemental information, and added a sentence in the main paper.

*"Figure 3 shows the comparison between observed and modeled hourly near surface air temperature, O$_3$ concentrations, and daily PM$_{2.5}$ concentrations. (Comparisons between observed and modeled diurnal cycles for near surface air temperatures and O$_3$ concentrations are also presented in the supplemental information, Figure S11 and S12.) As shown in Figure 3 (and Figure S11), the model simulations better capture higher air temperatures during the daytime relative to lower values during nighttime. By contrast, predictions of O$_3$ and PM$_{2.5}$ concentrations show good fit with observations at low values that occur with high occurrence frequency. However, observed O$_3$ and PM$_{2.5}$ concentrations are underestimated by the model at higher values that occur with lower frequency of occurrence."*

[Figure]

**Figure S11** Diurnal cycle of observed and modeled near surface air temperature.

[Figure]

**Figure S12** Diurnal cycle of observed and modeled surface O$_3$ concentrations (ppm).

6. Figs 5, 7 and 9 include values of simulated fields within urban grid cells only. The authors should consider superimposing in these figures values for the entire domain including nonurban grid cells. It would be very helpful to see the differences in the simulated fields within both urbanized pixels and also grid cells that remain natural in both scenarios considered.

This is a good idea. We added new versions of each figure to the supplemental information that include values for non-urban cells (Figures S13, S17 and S18). Please check the supplemental information for these three new figures. In general, the changes in non-urban grid cells are not significantly different from zero at 95% confidence interval for most places. We also added several sentences to the main paper which point to those figures.

Last sentence in section 3.2.2

*"A modified version of Figure 5 that includes values for non-urban cells is in the supplemental information Figure S13."*

Last sentence in section 3.3.1

*"A modified version of Figure 7 that includes values for non-urban cells is in the supplemental information Figure S17."*

Last sentence in section 3.4.2

*"A modified version of Figure 9 that includes values for non-urban cells is in the supplemental information Figure S18."*

---

## Referee Report (RR1)

Dear Editor,

The Authors have addressed most of the questions in an acceptable way and have revised the manuscript accordingly. Still I am not fully convinced about the terms 'historical urbanization' or 'prior to human settlement'. Clearly it is a sensitivity study on the effect of urbanization (roughness elements, excessive heat) on boundary layer structure (dynamical, thermal) which in turn feedbacks on chemical reactions and chemical transport processes.
Using identical emissions for different model scenarios further does not relate to a 'true' development but highlight more or less an idealized analysis which tries to separate chemical and dynamical effects. A minor point here is, that 'shrubland' is not supposed to be the dominant land use from 'before human settlement' I think, but more a combination of wetland/grassland/shrubland however. Maybe try to just highlight more prominently that the sensitivity study purely treats the conversion from 'real' urban morphology to shrubland for the urban area of LA. In this course I would also recommend to change the term 'present day' to 'Urban'.

Please use the term WRF-Chem instead of WRF/Chem

Compared to your past work, what is the benefit of including the 'shadow model'

*Question1: The scope of the study…*
Please further highlight your conclusion about the 'most important land surface factors'

*Question 2: irrigation module*
I still do not get the benefit of using the irrigation module and not discussing the sensitivity on the model results.

Question 3: Past model exercises have shown, that including the multi-layer UCM rather than the single layer model is not per-se a more complex exercise, as it basically just involves changing a switch in the namelist.input. In my understanding, the urban canopy model is further not coupled directly to WRF-Chem but to the (Noah) land-surface model and the link to air chemistry works over the changed atmospheric dynamics. It is correct though, that only a couple of boundary layer schemes do work with BEP and the vertical levels have to be adapted, but besides the potential higher computational costs, there should not be an enormous amount of extra work. Please comment briefly on that in the revision, or leave out some of the statements.

*Question 10: introduction*
How did recent studies make it possible to utilize 'satellite land surface data'. What is meant by the latter term?

Comment Line 86:
still unclear about the term 'meteorology can affect emission rates'

Comment to Line 119:
The way, the urban canopy model treats the land surface is not different to other existing studies. The mean values of the various parameters for a grid cell classified as 31,32,33 however are calculated from a potentially higher resolution and state of the art input data set. Maybe I am wrong here, but as you are not considering the tile-approach, the UCM essentially should treat an urban grid cell also as 'homogenous'. I think it is fair to say something like 'the urban canopy model is configured for the urban area of Los Angeles, whereas the values in each urban class are calculated including high

resolution land surface/building data. Not sure though if the impact of this high resolution data set on the final result differs much compared to a more simple urban canopy parametrization and what would be the effect on the difference between Urban-NoUrban.

I would recommend to better clarify these points in a revised version in the methodology and discussion/conclusion section.

*Abstract*
change the term 'before human settlement' here

*Underestimation of PM 2.5:*
The main reason for underestimation of the PM2.5 concentration in my opinion is the vertical resolution of the model in combination with the problem in comparing point with grid cell. Much of the pollutants measured at the surface might well be diluted by vertical mixing within the first 10s of meters. Can you please present this discussion more clearly, stating further if the particularly poor correlation has the same reasons. I can understand that these systems are not able to picture near surface concentrations, but however you should try to highlight better what are the main points you are interested in (e.g relative differences instead of absolute values).

---

## Author Response (AR2)

**Response to Anonymous Referee #2 (Report #1)**
(Note: Reviewer comments are listed in grey, and responses to reviewer comments are in black. Pasted text from the new version of the paper is in italics.)

Dear Editor,

The Authors have addressed most of the questions in an acceptable way and have revised the manuscript accordingly. Still I am not fully convinced about the terms 'historical urbanization' or 'prior to human settlement'. Clearly it is a sensitivity study on the effect of urbanization (roughness elements, excessive heat) on boundary layer structure (dynamical, thermal) which in turn feedbacks on chemical reactions and chemical transport processes.
Using identical emissions for different model scenarios further does not relate to a 'true' development but highlight more or less an idealized analysis which tries to separate chemical and dynamical effects. A minor point here is, that 'shrubland' is not supposed to be the dominant land use from 'before human settlement' I think, but more a combination of wetland/grassland/shrubland however. Maybe try to just highlight more prominently that the sensitivity study purely treats the conversion from 'real' urban morphology to shrubland for the urban area of LA. In this course I would also recommend to change the term 'present day' to 'Urban'.

We thank the reviewer for the comments and suggestions. Our focus is on land cover changes from "before human settlement" to "present-day". To make this clear, the abstract states,

*"We assume identical anthropogenic emissions for the simulations carried out, and thus focus on the effect of changes in land surface physical properties and land surface processes on air quality."*

Shrubland is a reasonable assumption for the dominant land use type for 'prior to human settlement' because 1) most of current rural area within Southern California is shrubland, and 2) the major climate types of Southern California include Mediterranean, semi-arid and desert with infrequent rain. The chance of growing a large area of grassland or wetland is low in this region.

Please use the term WRF-Chem instead of WRF/Chem

Thanks for the suggestion. We went through the main paper and supplemental information, and changed all 'WRF/Chem' to WRF-Chem.

Compared to your past work, what is the benefit of including the 'shadow model'

We included the shadow model because it tracks direct and diffuse radiation within the urban canopy separately, whereas with the shadow model turned off, all shortwave radiation within the urban canopy is assumed diffuse. Physically speaking, the first parameterization is more appropriate compared the second one, thus we turned on the

shadow model. This is discussed already in the last paragraph in section 3.2.3.

*Question1: The scope of the study…*
Please further highlight your conclusion about the 'most important land surface factors'

We modified the last paragraph in the conclusion section, and added how the important land surface factors drive changes in regional meteorology and air quality.

*"We find that increases in evapotranspiration, thermal inertia, and surface roughness due to historical urbanization are the main drivers of regional meteorology and air quality changes in Southern California. During the day, our simulations suggest that increases in evapotranspiration and thermal inertial from urbanization lead to regional air temperature reductions. Temperature reductions together with increases in surface roughness contribute to decreases in ventilation and consequent increases in ozone and PM$_{2.5}$ concentrations. During nighttime, increases in thermal inertial from urbanization lead to increases in regional air temperatures. Increases in temperatures together with increase in surface roughness lead to decreases in NOx and PM$_{2.5}$ concentrations. O$_3$ concentrations increase because of decreased titration by NOx."*

*Question 2: irrigation module*
I still do not get the benefit of using the irrigation module and not discussing the sensitivity on the model results.

The benefit of using the irrigation module is discussed in the last paragraph of section 2.2 in the main paper. Using the irrigation module can improve the model performance in predicting latent heat fluxes and temperatures for the Los Angeles region. In other words, using the irrigation model allows for a more realistic "present-day" scenario. (Please refer to Vahmani and Hogue (2014) for details on the implementation and evaluation of the irrigation module.) Using the irrigation module thus gives us a more reliable result on the effect of land surface changes via urbanization on regional meteorology compared to not involving the irrigation module.

We have also discussed the effect of land surface changes via urbanization without involving irrigation in supplemental information section S3. By comparing the result in section 3.2.1 and 3.2.2 in the main paper with section S3 in supplemental information, we can get the sensitivity of turning on/off irrigation module on the model results. Lastly, a previous paper by our group (Vahmani and Ban-Weiss, 2016a) includes sensitivity simulations with and without irrigation.

Question 3: Past model exercises have shown, that including the multi-layer UCM rather than the single layer model is not per-se a more complex exercise, as it basically just involves changing a switch in the namelist.input. In my understanding, the urban canopy model is further not coupled directly to WRF-Chem but to the (Noah) land-surface model and the link to air chemistry works over the changed atmospheric dynamics. It is correct though, that only a couple of boundary layer schemes do work with BEP and the vertical

levels have to be adapted, but besides the potential higher computational costs, there should not be an enormous amount of extra work. Please comment briefly on that in the revision, or leave out some of the statements.

We agree with the reviewer that the major concern with using BEP is that the resulting model would be potentially more computationally expensive, but for likely little additional benefit in the quality of simulations. Adding the chemistry module already adds immense computational cost. We added a sentence in the first paragraph in section 2.1 to comment on why we used UCM instead of BEP.

*"We used UCM instead of the multilayer canopy layer model (BEP) because BEP would increase computational costs, but for likely little additional benefit in the quality of simulations (Chen et al., 2011; Kusaka et al., 2001)."*

*Question 10: introduction*
How did recent studies make it possible to utilize 'satellite land surface data'. What is meant by the latter term?

Satellite-retrieved land surface data are those land surface data (e.g., surface albedo, vegetation fraction, leaf area index, etc.) from remote sensing instruments (e.g., MODIS that we used) onboard satellites. Raw satellite-retrieved land surface data cannot be used directly as input in default WRF. Therefore, some recent studies incorporated these data to WRF modeling by adjusting data format and modifying related model code. These studies also assessed the improvements in model performance using land surface data from satellites compared to default data available in the WRF package. Please refer to our previous paper Vahmani and Ban-Weiss, (2016b) as an example of this type of study.

Comment Line 86:
still unclear about the term 'meteorology can affect emission rates'

Meteorological conditions such as temperature and sunlight intensity can affect rates of biogenic emissions and also evaporative emissions of fuels like gasoline. We added the word 'biogenic' to the sentence.

*"Meteorology can affect emission rates (e.g., biogenic volatile organic compounds (BVOCs) and evaporative emissions of gasoline), chemical reaction rates, gas-particle phase partitioning of semi-volatile species, pollutant dispersion, and deposition…"*

Comment to Line 119:
The way, the urban canopy model treats the land surface is not different to other existing studies. The mean values of the various parameters for a grid cell classified as 31,32,33 however are calculated from a potentially higher resolution and state of the art input data set. Maybe I am wrong here, but as you are not considering the tile-approach, the UCM essentially should treat an urban grid cell also as 'homogenous'. I think it is fair to say something like 'the urban canopy model is configured for the urban area of Los Angeles, whereas the values in each urban class are calculated including high resolution land

surface/building data'. Not sure though if the impact of this high resolution data set on the final result differs much compared to a more simple urban canopy parametrization and what would be the effect on the difference between Urban-NoUrban.

We are not saying that each urban grid cell has wide heterogeneity, but instead saying that the urban land surface properties for the whole modeling domain shows high spatial heterogeneity. Most past studies use unvarying table-values for land surface properties based on land cover types; thus, properties for any urban cell would be unvarying. We added the word 'spatial' to the related sentence.

*"They also do not resolve the wide spatial heterogeneity of urban land surface properties, with most studies assuming that urban properties are homogenous throughout the city."*

By resolving the real-world spatial heterogeneity in WRF, the model performance of predicting temperature is improved. Our previous research Vahmani and Ban-Weiss (2016b) has shown that using land properties from MODIS reduces model biases in surface and near-surface air temperatures (relative to ground and satellite observations) for urban regions in southern California compared to using default datasets available with WRF. In particular, the root-mean-square-error for nighttime near-surface air temperature has been narrowed from 3.8 to 1.9 °C.

I would recommend to better clarify these points in a revised version in the methodology and discussion/conclusion section.

Ok, please see all responses above.

*Abstract*
Change the term 'before human settlement' here

We think 'before human settlement' is clear to the readers, so we keep the text as it is.

*Underestimation of $PM_{2.5}$:*
The main reason for underestimation of the $PM_{2.5}$ concentration in my opinion is the vertical resolution of the model in combination with the problem in comparing point with grid cell. Much of the pollutants measured at the surface might well be diluted by vertical mixing within the first 10s of meters. Can you please present this discussion more clearly, stating further if the particularly poor correlation has the same reasons. I can understand that these systems are not able to picture near surface concentrations, but however you should try to highlight better what are the main points you are interested in (e.g relative differences instead of absolute values).

Thanks to the reviewer for this suggestion. It is a very good point. We added this reason to our explanation of the underestimation of $PM_{2.5}$ concentrations, and removed several minor points. We also emphasized that we are interested more in relative differences rather than absolute values. The modified sentences are as follows.

*"The underestimation of PM$_{2.5}$ concentrations may be occurring mainly due to the following factors: 1) not including dust emissions in the simulation, which makes up an appreciable fraction of real-world total PM$_{2.5}$, and 2) while the observations measure values for one single point near the surface, model values represent a grid cell average with a larger spatial "footprint". Note that the focus of this study is on the changes in pollutant concentrations, and thus relative differences are of increased interest relative to absolute values."*

**Response to Anonymous Referee #3 (Report #2)**
(Note: Reviewer comments are listed in grey, and responses to reviewer comments are in black. Pasted text from the new version of the paper is in italics.)

The revised manuscript is significantly improved compared to the previous version. Many of my comments have been addressed. I have few comments and a concern regarding the interpretation of the statistical significance of the land surface changes impacts on regional meteorology and air quality; and the variable used to derive the confidence levels.

We thank the reviewer for the comments and suggestions. Please find our point-to-point responses as follows.

Major comment:

The authors state in many places of the manuscript "significantly differ from zero at the 95% confidence level" (i.e. line 355, 425, 435, and elsewhere). The measure used for the statistical significance is not meaningful since the differences in the fields simulated under the scenarios considered are likely expected to be different from zero. In fact, even WRF simulations using same configurations but different computer platforms or compiler options can lead to different results due to errors. I would suggest basing the statistical significance on the sign of the difference in the fields simulated. The confidence level at which the difference is positive (negative) if you are expecting increases (decreases) would be more meaningful.

We believe that our current way of carrying out the statistical test is reasonable for the following reasons. Firstly, we used one computing platform and model configuration for all simulations, and thus this cannot be a cause for variability. So if the comparison between two scenarios significantly differs from zero, it is the different settings between these two scenarios that cause the difference. Secondly, different settings between two scenarios does not necessarily mean significant difference in model results if the differences are smaller than model noise. Thus, it is meaningful to check the statistical significance between two model scenarios. Last but not least, this type of statistical test is widely used to study whether changes are significant (Vahmani et al., 2016; Zhang et al., 2016). Using your recommendation to assess significance based only on whether the difference has the same sign as expectation is circular; there is no "expected" sign of change given that many competing pathways contribute to modeled changes. Plus, this would not be an objective approach, only deeming things significant if they follow our expectation.

Other comments:

1- Please indicate the year of the JJA observations discussed in the text (Line 180) and in the caption of Figure S8.

Thanks for catching it. We added 'year 2012' in both two places.

2- Line 310: Please use a different word for "validation". I would suggest "verification". A complex model like WRF cannot be validated. Verifications of WRF are always required as the WRF results give different results depending on the configurations, regions, periods and the physics parameterizations employed.

Thanks for pointing it out. We changed 'validation' to 'verification' in the paper and supplemental information.

3- Line 290 and elsewhere: Please reconsider the interpretation of the statistical significance (see the comment above).

Please refer to our response to the major comment.

4- Line 345: I would remove the sentence "Given that the atmosphere is stratified in models". The discrete approximation shown in Eq2 does not depend on the stratification of the atmosphere since you are not using temperature or density as an independent variable in the integral. I would remove Eq2 (or Eq1) since they are equivalent.

Thanks for the suggestion. We removed Eq1 (since Eq2 is what we actually used to calculate ventilation coefficient) from the paper.

*"...This calculation can be written as (Eq1).*

$$Ventilation\ Coefficient = \ \sum_{i=1}^{m} U(z_i) \times \Delta z_i \qquad (Eq1)$$

*where $U(z_i)$ stands for horizontal wind speed within the $i^{th}$ model layer (m/s), $\Delta z_i$ is the depth of the $i^{th}$ model layer that is within the PBL (m), and m is the number of vertical layers up to PBL height."*

[revised manuscript text omitted]